# FreshStack: Building Realistic Benchmarks for Evaluating Retrieval on Technical Documents

**Nandan Thakur** [1][*]     **Jimmy Lin** [1]     **Sam Havens** [2]
**Michael Carbin** [2]     **Omar Khattab** [2]     **Andrew Drozdov** [2]
[1]University of Waterloo, Canada     [2]Databricks, USA
https://fresh-stack.github.io

## Abstract

We introduce FreshStack, a holistic framework for automatically building information retrieval (IR) evaluation benchmarks by incorporating challenging questions and answers. FreshStack conducts the following steps: (1) automatic corpus collection from code and technical documentation, (2) nugget generation from community-asked questions and answers, and (3) nugget-level support, retrieving documents using a fusion of retrieval techniques and hybrid architectures. We use FreshStack to build five datasets on fast-growing, recent, and niche domains to ensure the tasks are sufficiently challenging. On FreshStack, existing retrieval models, when applied out-of-the-box, significantly underperform oracle approaches on all five domains, denoting plenty of headroom to improve IR quality. In addition, we identify cases where rerankers do not improve first-stage retrieval accuracy (two out of five domains) and oracle context helps an LLM generator generate a high-quality RAG answer. We hope FreshStack will facilitate future work toward constructing realistic, scalable, and uncontaminated IR and RAG evaluation benchmarks.

## 1 Introduction

Retrieval-augmented generation (RAG) is a popular technique to enhance traditional information retrieval (IR) capabilities with language model generation. RAG systems use large language models (LLMs) to generate long-form responses [22, 37, 25, 4], grounded in the information available from retrieved documents [33, 37, 20, 45]. Despite its wide usage, evaluating RAG remains incredibly challenging. Existing IR and RAG benchmarks are not well-suited for evaluation, as these are outdated and highly limited. In particular, we observe three major issues in existing benchmarks:

- **Lack of realistic, open-ended questions**: Existing datasets contain purely extractive short answers (e.g., Natural Questions [36], TriviaQA [30]) or crowd-sourced questions (e.g., HotPotQA [88]). A limited number of datasets capture "natural" human-asked questions, i.e., MS MARCO [50] or Natural Questions [36], but unfortunately, brief and straightforward questions are inserted into a search box, failing to represent the complex questions that real users might pose to RAG systems.

- **Artificially easy**: RAG represents an *approach* rather than a *problem*. Real users require systems capable of grounded question answering, i.e., responding to specialized questions by referencing knowledge from a document corpus. Consequently, datasets constructed by design to be solvable via retrieval often fail to encode challenges faced in RAG applications.

- **Static and unspecialized**: After sourcing questions and answers, a benchmark becomes at the risk of (1) *contamination*, if current LLMs are trained on the same set of documents or questions, (2) *overfitting*, when systems inevitably saturate by repeated internal or external leaderboarding (e.g., BEIR [74]), and (3) *staleness*, when questions or answers are not refreshed and become outdated.

---

[*]Work done during Nandan's internship at Databricks.

39th Conference on Neural Information Processing Systems (NeurIPS 2025) Track on Datasets and Benchmarks.

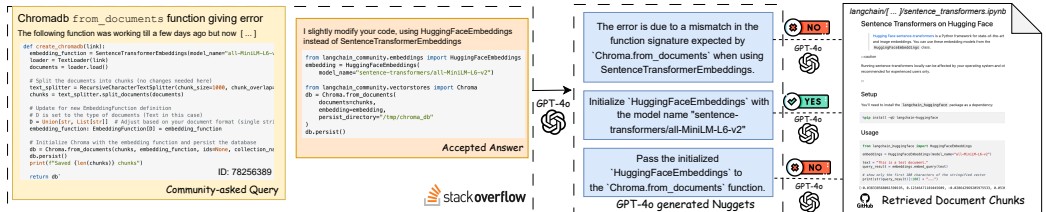

Figure 1: A data instance from LangChain generated with FreshStack. The question and answer pair is sourced from Stack Overflow. The pair is provided to GPT-4o to generate nuggets, highlighting necessary facts in the answer. Next, code snippets and technical documents from multiple GitHub repositories (e.g., Jupyter Notebook) are chunked, processed, and pooled for each question. Finally, each pooled document chunk is judged with GPT-4o for binary relevance (either yes or no) at a nugget-level, i.e., whether the document factually supports the information present in each nugget.

A realistic benchmark should measure model generalization on niche domains, and continue to update. Additionally, it must capture the complexity of human-generated queries—such as multi-hop reasoning [88], code understanding [29], or specialized terminology—rather than relying on artificially easy questions. This drives the robustness of systems in answering questions in evolving public libraries or private code bases [92, 29], a company's internal forum [60], or technical troubleshooting [69, 58, 8].

In our work, we introduce **FreshStack**, a holistic framework for constructing realistic datasets on niche and challenging domains, seeking to avoid contamination due to (perpetual) recency. Using FreshStack, we construct an evaluation benchmark on five niche domains sourced from community-asked questions and answers on Stack Overflow and a corpus containing code snippets and technical documents from public GitHub repositories. The framework contains three major steps: (1) *automatic corpus collection* (Section 3.2) with technical documents chunked and sourced from several GitHub repositories, (2) *nugget generation* (Section 3.3) with GPT-4o using community-asked questions and answers in Stack Overflow, and (3) *nugget-level support* (Section 3.5) with GPT-4o on question and document chunks, retrieved from judgment pools constructed using a fusion of retrieval techniques.

We investigate three research questions in our work to provide insights on FreshStack: **RQ1** How to construct challenging evaluation datasets from real user-asked questions? **RQ2** How do LLMs act as an assessor for nugget generation on community-asked questions & answers and nugget-level support with retrieved documents? **RQ3** How do state-of-the-art retrieval models, rerankers, and LLM generators perform on IR and RAG evaluation benchmarks generated with FreshStack?

We calibrate the automatic stage in FreshStack with GPT-4o using a machine learning (ML) expert, assessing the quality of nugget generation and nugget-level support for one of the domains (LangChain). Our results show that GPT-4o-generated nuggets capture crucial information required to answer the question, and GPT-4o precisely labels support at a nugget level. For pooling, we compare oracle (having access to the answer) and inference (relying only on the question) settings, finding that question decomposition and nugget generation outperform other techniques, respectively.

Beyond pool construction, we evaluate retrieval and rerankers in the document retrieval setting using only the Stack Overflow question. Retrieval models drastically underperform oracle systems on all five domains, showing a high headroom for improvement. In addition, ensemble fusion outperforms individual models, indicating that diversity in models enhances retrieval, and rerankers provide clear benefits in some but not all domains. Finally, we evaluate answer generation, where the oracle context assists the LLM generator to provide a high-quality RAG answer. FreshStack is a general framework and can be applied to any domain of a similar structure. Overall, we hope the framework serves as a testbed for future work to develop challenging benchmarks for evaluating RAG systems.

## 2 Related Work

**Retrieval-augmented generation.** RAG has been widely used to avoid "hallucinations" [94] seen with LLMs when handling knowledge-intensive tasks [31]. RAG reduces factually incorrect generated content, leading to adoption in various commercial systems, e.g., Bing Search or Google AI Overviews. Existing IR and RAG benchmarks are stale, evaluating on academic question answering datasets [20, 63, 61, 71, 86], or are not challenging, being constructed for RAG [87, 7, 40, 51, 71, 34]. A limited number of datasets refresh over time to avoid LLM decontamination [32, 80, 67, 84],

however, these contain easy and unrealistic questions. In contrast, FreshStack generates niche and challenging datasets, which can refresh over time and are not constructed specifically for RAG.

**Code-based benchmarks.** Neural code generation [47] requires LLMs to generate code from scratch for generic programming questions. One popular benchmark is SWE-Bench [29], which evaluates whether LLMs can generate code changes for GitHub pull requests (PRs) in popular public repositories. Similarly, CodeSearchNet [24], COIR [39], LiveCodeBench [27], and CodeRAG-Bench [82] focus on the evaluation of high-level programming problems on popular public repositories. In contrast, in FreshStack, we focus on assisting developers, from a novice to a domain expert, by providing real-time answers on fast-growing and recent domains such as LangChain (introduced in 2023) by referencing technical documentation in GitHub repositories.

**Stack Overflow datasets.** FreshStack is *not* the first dataset to leverage Stack Overflow for retrieval, the evaluation setting of retrieving canonical documents from GitHub repositories remains unexplored. Existing datasets such as CQADupstack [23], LoTTE [65], and Stack Overflow-QA [39] address a different task, to retrieve the relevant answer snippet in response to a real user question on Stack Overflow. The closest setting is Neural Code Search [38], which contains simple questions to retrieve code snippets from GitHub on popular programming domains, such as Android. In contrast, FreshStack contains multifaceted and complex queries on niche domains, such as LangChain.

## 3 The FreshStack Framework

The framework involves five major stages to construct an evaluation dataset (as highlighted in Figure 2). FreshStack includes three major design choices:

1. A general framework that can be extended to different domains without manual effort.
2. Adding recent and niche domains actively discussed in computer programmer communities such as Stack Overflow.
3. Sourcing community-asked questions and answers, to make our evaluation challenging, requiring domain expert knowledge to answer them correctly.

**Stack Overflow** is an online question answering platform for computer programmers. Users ask questions about a particular tag and provide a description (often a code snippet with the error message) with Stack Overflow tags. Questions and answers are also tagged, allowing for easy retrieval of tag or domain-wise questions.

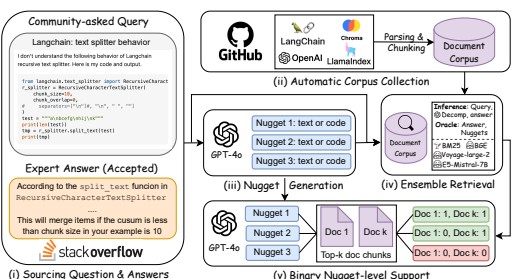

Figure 2: The FreshStack framework: (1) Stack Overflow questions and answers are sourced for recent and niche domains. (2) GitHub repository documents are collected and chunked to form the corpus (for each domain). (3) Nuggets or atomic facts within the question and answer are generated with GPT-4o. (4) Ensemble techniques and models retrieve documents, which construct our document judgment pools. (5) GPT-4o evaluates support for every document-nugget pair as a binary judgment.

### 3.1 Stack Overflow Domain and Question Selection

For FreshStack, we target niche and recent domains (or topics) introduced on Stack Overflow from 2023 onward (frequency in Figure 3), containing a minimum of 50 posts. We sort all domains using the overall number of posts and curate five domains starting from the highest (LangChain) to the lowest frequency (Yolo v7 & v8), covering different areas and are sufficiently different from each other. Each area represents a unique domain, e.g., Computer Vision (CV) or Machine Learning (ML).

**Questions & Answers.** We extract relevant posts and answers from the Stack Overflow XML data archive (dated October 2024).[2] We scan the archive to pick all the relevant posts as questions containing the required tag and further filter answer posts to the questions. Finally, we keep questions with an *accepted answer*, prioritizing precision over quantity in retaining questions with only high-quality answers accepted and upvoted in the Stack Overflow community.

---

[2]The Stack Overflow XML data archive (CC BY-SA license) is updated once every quarter: https://meta.stackexchange.com/questions/401324/announcing-a-change-to-the-data-dump-process

## 3.2 Automatic Corpus Collection with GitHub Technical Documents

Answering a higher percentage of questions requires a robust set of corpora from *multiple sources*. For instance, addressing issues in LangChain may require ChromaDB GitHub documentation to resolve errors related to its usage. In our work, we build a different document corpus per domain by combining multiple repositories as sources (we list each repository per domain in Table 8).

**Stack Overflow Tags.** We analyze each tag frequency from Stack Overflow to select a relevant GitHub repository to be included in the document corpus. This involves identifying top-k co-occurring tags, where k is the threshold balancing question coverage with indexing costs. Some tags are generic, such as Python, whereas others are specific, such as LangChainJS. Filtering the tags to keep only a subset of repositories for a domain does not degrade the dataset quality. We manually verify each GitHub repository for each tag, with plans to automate this procedure in the future.

**Chunking & Indexing.** We clone the latest branch of the GitHub repository (on 22nd October 2024) and parse files as a tree structure. Each file (either a text document or code snippet) is chunked into small sections containing a maximum of **2048 tokens**, skipping non-text formats.[3] The GitHub filepath serves as the document identifier, with additional chunk details encoded in the identifier. Finally, we combine all chunks into a single corpus, prefixing all document identifiers with the repository name to uniquely identify the common files separately (e.g., LICENSE or requirements.txt).

## 3.3 Nugget Generation with Stack Overflow Question & Answer

A nugget is a core concept or an atomic fact essential in a system's response. The term nugget was informally referred to as SCU (summary content units) as clauses appearing in model summarization [49] and later formalized as "information nugget" for evaluating long-form answers [79, 42, 43, 54]. *Nugget generation* refers to the procedure of constructing nuggets from information-dense text. The procedure decomposes a verbose answer into key atomic facts or essential components, aiding evaluation. The nugget-generation methodology was first coined two decades ago in the TREC-QA 2003 track for evaluating answers to free-form questions [79]. Back then, human annotators would manually write "information nuggets" or atomic facts required to be present in the answer. More recently, with the onset of RAG, nugget-based evaluation has renewed interest with LLMs for factual accuracy assessment in long answers, where automatic nugget generation is explored, i.e., using LLMs to automatically create nuggets [2, 48, 18, 55–57].

**Nugget Generation Setting.** We automatically generate nuggets from Stack Overflow question-answer pairs using GPT-4o [52], avoiding the cumbersome procedure of manual nugget construction [55]. LLM-based nugget generation has been explored in the TREC 2024 RAG track [55–57] and in multiple works [15, 18]. Separately, we experimented with prompting techniques and found that grading notes style prompts [46] provided parseable and high-quality nuggets in our experiments. An example of GPT-4o-generated nuggets for a question in LangChain is shown in Table 9.

## 3.4 Retrieval: Oracle & Inference Setting

A RAG evaluation dataset requires questions, answers, and a corpus with documents, which helps support crucial facts present in the answer. Pooling [95, 5] is a predominant technique used in IR for selecting a subset of documents to be assessed for relevance, instead of assessing every document from the collection. We retrieve a list of highly relevant (unjudged) documents from the corpus and construct judgment pools. Since, we are constructing an *evaluation dataset* and we have *curated answers* for questions, we retrieve documents using two methods: (1) **Inference**, relying only on the question and automatic approaches, and (2) **Oracle**, relying on the gold answer or list of nuggets, to construct judgment pools for the support (or relevance) judgment task in the next stage.

**Retrieval Setting.** We experiment with multiple systems to increase diversity in our judgment pools. First, we experiment with two techniques in the *inference setting*: (i) GPT-4o Sub-Questions: we decompose the original question and generate synthetic sub-questions with GPT-4o, similar to Rosset et al. [64], concatenated together to retrieve documents, and (ii) GPT-4o Closed Book Answer: we generate a closed-book answer for the original question with GPT-4o, similar to HyDE [19], and use the closed-book answer to retrieve documents. Next, in the *oracle setting*, we experiment with:

---

[3]We skip indexing images, videos, .bin, .csv, and audio files or unrecognized file formats.

| Domain | Area | Dataset Count | | | | Avg. Length | | % Containing Code | | Relevance Judgments | |
|---|---|---|---|---|---|---|---|---|---|---|---|
| | | #Q | #Docs | #GH | Avg. N/Q | Query | Answer | Query | Answer | Rel. Docs/N | Rel. Docs/Q |
| **LangChain** | Machine Learning (ML) | 203 | 49,514 | 10 | 3.1 | 473.4 | 233.8 | 83.3% | 62.1% | 5.7 | 10.9 |
| **Yolo v7 & v8** | Computer Vision (CV) | 57 | 27,207 | 5 | 3.5 | 497.1 | 191.7 | 70.2% | 71.9% | 3.9 | 7.4 |
| **Laravel 10 & 11** | Backend Development | 184 | 52,351 | 9 | 3.0 | 474.4 | 155.5 | 43.5% | 51.1% | 3.2 | 6.0 |
| **Angular 16, 17 & 18** | Front-end Development | 129 | 117,288 | 4 | 3.2 | 463.3 | 215.1 | 69.8% | 57.4% | 4.4 | 8.7 |
| **Godot4** | Game Development | 99 | 25,482 | 6 | 3.3 | 350.4 | 263.0 | 52.5% | 52.5% | 2.9 | 5.9 |

Table 1: FreshStack dataset statistics; Dataset count measures the number of queries, documents, GitHub repositories, and average nuggets per query; Avg. length measures the average text lengths (length distribution in Figure 4); % containing code measures the fraction of queries and answers with code snippets; Relevance judgments measure relevant documents per nugget and per query.

(i) Stack Overflow Answer: we use the curated Stack Overflow answer as the question to retrieve documents, and (ii) Stack Overflow Nuggets: we use the list of GPT-4o-generated nuggets (Section 3.3), concatenated as the question to retrieve documents.

**Retrieval Models.** We experiment with five different code and text-aware retrieval models: (i) **BM25**, a strong lexical baseline in BEIR [74]. We utilize the default implementation available in Pyserini [44]. (ii) **BGE (Gemma-2)** [7] a dense retriever model[4] fine-tuned on the backbone architecture of Gemma-2 (9B) [62] with an embedding size of 3584 and 8K context length. (iii) **E5 Mistral (7B)** [81] is a dense retriever model[5] based fine-tuned on the backbone of Mistral 7B [28] with 32 layers and embedding size of 4096. (iv) **Voyage-large-2**[6] is a proprietary and general-purpose embedding model optimized for retrieval quality, with a context length of 16K tokens and embedding size of 1536. (v) **Ensemble Fusion** is defined as the process of combining multiple search techniques (or models) to increase the overall relevance and accuracy of retrieved results [35]. We use a hybrid retrieval strategy combining all four models (i–iv), normalizing and summing up their individual similarity scores for a maximum of 100 documents for each model.

### 3.5 Nugget-Level Support Assessment with Retrieved Documents

Traditionally, relevance judgments are conducted on selected pools of retrieved documents, i.e., where a human assessor judges the relevance of the question with each provided document. Due to computational costs, recent studies experiment with an LLM judge (instead of a human assessor) for relevance judgments in IR [17, 75, 77, 76, 59]. Questions in existing IR datasets are traditionally short, making it easier to judge document relevance. In contrast, questions in the FreshStack dataset are long and elaborate (between 350–500 tokens in length), containing a mixture of text, code snippets, or outputs, making it challenging to judge question-document relevance directly [14]. For instance, a document may answer a major problem presented in the question, address only part of the question, or contain relevant references and examples, and we need to translate this into a relevance score.

**Nugget-level Support Setting.** Instead of relying on traditional relevance assessments, we simplify the judgment procedure for GPT-4o and evaluate whether a document supports information (or contains) provided by a nugget.[7] A reminder that a nugget highlights an essential fact of the Stack Overflow question or answer. Judging document relevance at a nugget level is effective as nuggets are factual and short information snippets, reducing the ambiguity often seen during traditional relevance judgments. To reduce computational costs, we evaluate top-k documents (a maximum $k = 20$) together with the list of all nuggets for a question in a single inference call $(n + k)$. We evaluate support judgment with GPT-4o using a chain-of-thought prompt [83].

## 4 Dataset Statistics & Evaluation

Completing previous stages, we employ two additional post-processing steps to ensure high-quality question and answer pairs remain in the dataset, sacrificing the overall dataset size. In the first step, we remove unsupported questions, i.e., questions that do not contain even a single relevant document;

---

[4]BGE Gemma-2: https://huggingface.co/BAAI/bge-multilingual-gemma2

[5]E5 Mistral 7B: https://huggingface.co/intfloat/e5-mistral-7b-instruct

[6]Voyage-large-2: https://docs.voyageai.com/docs/embeddings

[7]This is analogous to relevance judgment in traditional IR, but we effectively coined it "support" to calculate whether the document can sufficiently support the information present in the nugget, instead of relevance.

this removes, on average, **11.8%** of the total questions.[8] In the next step, we aggressively filter by removing questions containing at least one unsupported nugget, i.e., a nugget not supported by any documents, reducing on average **34.2%** of the total questions.

## 4.1 Dataset Statistics

FreshStack datasets covers five domains for programmers: machine learning, computer vision, backend, front-end, and game development, all listed in Table 1. Stack Overflow domains such as LangChain were introduced in 2023, whereas others, like Laravel or Angular, have questions about the latest versions (e.g., Laravel 10 & 11). Each domain has at least 50 questions, all asked between January 2023 and June 2024 (timeline versus frequency shown in Figure 3). The corpus has at least 25K documents sourced from 4–10 GitHub repositories (repositories listed in Table 8). The questions are even longer than answers (distribution shown in Figure 4), containing 350–500 tokens (computed using GPT-4o tokenizer), and at least 50% of the questions and answers contain code snippets. GPT-4o generates around 3–4 nuggets for each question. Each nugget supports at least 3 relevant documents, resulting in 5–6 relevant documents per question, for all domains.

**Dataset Licensing, Ethics and PII.** FreshStack is released publicly under the CC-BY-SA license, matching Stack Overflow and compatible with the mixed licenses across GitHub repositories.[9] The individual GitHub repository licenses are listed in Table 8. We downloaded data in FreshStack through either the Stack Overflow data dump access link, or through each respective GitHub repository's git URL via git clone. We did not apply any additional PII-specific filtering beyond what was already done in the data sources.

## 4.2 Cost Comparsion: Automatic Pipeline vs. Human Judgments

The cost of automatic construction of FreshStack with GPT-4o is approximately 260-275 USD. Here is a rough estimate of the costs involved: (1) *nugget generation*: ($O((\alpha_i + \alpha_o) * n)$ cost, where $n$ is the number of queries) costs about 10–15 USD with GPT-4o to generate nuggets for all domains (2–3 USD per domain). (2) *nugget-level support*: ($O((\beta_i + \beta_o) * n * m)$ cost, where m is the depth of retrieved documents for each query) costs about 250–260 USD with GPT-4o to assess document support with query nuggets (40–75 USD per domain), where $\alpha$ and $\beta$ are constants mapping to expected token count, with subscripts $i$ and $o$ corresponding to input and output. So, overall our computational cost is around 260–275 USD for five domains.

In contrast, manually constructing FreshStack with human annotators is expensive. (1) *nugget generation*: There are 1,149 queries in total and assuming a human annotator requires 2–3 minutes to generate nuggets, requiring 50–60 hours to complete the task. (2) *nugget-level support*: Assuming at least 50 documents are labeled for each query, leading to 57,450 pairwise comparisons (query-document pairs). Assuming a single human annotator can annotate 1 pair in a minute (to read all the nuggets and document carefully and mark nuggets that are sufficiently supported by the document), completing all judgments would require roughly 950–1000 hours. Assuming annotators are paid 20 USD per hour, the cost of manually annotating FreshStack is around 19–20K USD.

## 4.3 Retrieval and RAG Evaluation Metrics

IR evaluation traditionally follows the Cranfield paradigm [78], focusing on individual document relevance, independent of other documents. This is used to construct standard test collections, such as BEIR [74] and TREC datasets such as the Deep Learning (DL) track [10, 11, 13]. However, diversity in search [6, 66, 85] penalizes information redundancy within retrieved documents to enrich information content and improve efficiency. Therefore, we evaluate retrieval systems with three metrics with relevance judgments at the nugget-level: $\alpha$-nDCG@10 for diversity and relevance, Coverage@20 for nugget coverage, and Recall@50 for traditional relevancy. For RAG evaluation, we compute the All Strict ($A_{strict}$) metric for nugget-based recall taken from TREC RAG 2024 [56, 57], calculating how many nuggets are supported within a system's response, where each nugget highlights a different aspect of the answer. Please refer to Appendix B for a detailed overview of each metric.

---

[8]Future work may include these questions as they are potentially valuable to answer, and better retrieval systems may be able to find relevant documents.

[9]Our dataset documentation states: "The original GitHub repositories used for constructing the corpus may contain non-permissive licenses; we advise the reader to check the licenses for each repository carefully."

| Method | Model | LangChain | | | Yolo v7 & v8 | | | Laravel 10 & 11 | | | Angular 16, 17 & 18 | | | Godot4 | | |
|---|---|---|---|---|---|---|---|---|---|---|---|---|---|---|---|---|
| | | αN@10 | C@20 | R@50 | αN@10 | C@20 | R@50 | αN@10 | C@20 | R@50 | αN@10 | C@20 | R@50 | αN@10 | C@20 | R@50 |
| *Inference Setting: Using a variant of the Stack Overflow question for retrieval of documents within the corpus* | | | | | | | | | | | | | | | | |
| GPT-4o Sub-Questions | BM25 | 0.228 | 0.495 | 0.249 | 0.150 | 0.427 | 0.328 | 0.349 | 0.656 | 0.464 | 0.307 | 0.666 | 0.378 | 0.154 | 0.326 | 0.211 |
| | BGE (Gemma-2) | 0.220 | 0.561 | 0.324 | 0.220 | 0.554 | 0.367 | 0.407 | 0.727 | 0.585 | 0.360 | 0.707 | 0.459 | 0.240 | 0.532 | 0.382 |
| | E5 Mistral (7B) | 0.262 | 0.613 | 0.362 | 0.266 | 0.593 | 0.484 | 0.306 | 0.643 | 0.528 | 0.305 | 0.617 | 0.397 | 0.220 | 0.461 | 0.349 |
| | Voyage-large-2 | 0.270 | 0.563 | 0.329 | 0.213 | 0.526 | 0.370 | 0.366 | 0.687 | 0.552 | 0.344 | 0.69 | 0.449 | 0.260 | 0.594 | 0.473 |
| | Fusion (4 models) | **0.322** | **0.708** | **0.475** | 0.305 | 0.665 | 0.489 | **0.478** | **0.763** | **0.662** | **0.428** | **0.817** | **0.584** | **0.290** | **0.598** | **0.526** |
| GPT-4o Closed Book Answer | BM25 | 0.256 | 0.520 | 0.273 | 0.286 | 0.554 | 0.431 | 0.376 | 0.655 | 0.495 | 0.293 | 0.542 | 0.332 | 0.241 | 0.473 | 0.349 |
| | BGE (Gemma-2) | 0.181 | 0.467 | 0.263 | 0.271 | 0.599 | 0.473 | 0.360 | 0.694 | 0.539 | 0.242 | 0.525 | 0.338 | 0.187 | 0.454 | 0.358 |
| | E5 Mistral (7B) | 0.198 | 0.471 | 0.277 | 0.239 | 0.511 | 0.364 | 0.188 | 0.458 | 0.384 | 0.179 | 0.430 | 0.267 | 0.151 | 0.318 | 0.237 |
| | Voyage-large-2 | 0.220 | 0.500 | 0.301 | 0.247 | 0.557 | 0.495 | 0.317 | 0.658 | 0.524 | 0.227 | 0.461 | 0.338 | 0.253 | 0.510 | 0.454 |
| | Fusion (4 models) | 0.275 | 0.630 | 0.432 | **0.356** | **0.686** | **0.578** | 0.420 | 0.738 | 0.641 | 0.290 | 0.582 | 0.470 | 0.288 | 0.538 | 0.492 |
| *Oracle Setting: Using the Stack Overflow answer directly or its variants for retrieval of documents within the corpus* | | | | | | | | | | | | | | | | |
| Stack Overflow Answer | BM25 | 0.461 | 0.726 | 0.428 | 0.481 | 0.756 | 0.574 | 0.511 | 0.774 | 0.588 | 0.469 | 0.751 | 0.521 | 0.325 | 0.565 | 0.397 |
| | BGE (Gemma-2) | 0.290 | 0.625 | 0.367 | 0.390 | 0.815 | 0.604 | 0.472 | 0.814 | 0.675 | 0.346 | 0.690 | 0.481 | 0.341 | 0.718 | 0.561 |
| | E5 Mistral (7B) | 0.331 | 0.671 | 0.430 | 0.315 | 0.683 | 0.509 | 0.260 | 0.634 | 0.488 | 0.291 | 0.570 | 0.412 | 0.277 | 0.546 | 0.434 |
| | Voyage-large-2 | 0.385 | 0.700 | 0.432 | 0.405 | 0.703 | 0.589 | 0.439 | 0.791 | 0.641 | 0.371 | 0.626 | 0.541 | | | |
| | Fusion (4 models) | 0.484 | 0.821 | 0.619 | 0.546 | 0.854 | 0.788 | 0.564 | **0.892** | **0.820** | 0.470 | 0.805 | 0.695 | 0.449 | 0.741 | 0.683 |
| Stack Overflow Nuggets | BM25 | 0.467 | 0.739 | 0.445 | 0.519 | 0.796 | 0.657 | 0.540 | 0.840 | 0.654 | 0.485 | 0.787 | 0.536 | 0.428 | 0.680 | 0.489 |
| | BGE (Gemma-2) | 0.308 | 0.667 | 0.405 | 0.461 | 0.784 | 0.572 | 0.448 | 0.806 | 0.666 | 0.393 | 0.756 | 0.536 | 0.335 | 0.664 | 0.555 |
| | E5 Mistral (7B) | 0.323 | 0.684 | 0.432 | 0.437 | 0.737 | 0.554 | 0.287 | 0.631 | 0.533 | 0.346 | 0.670 | 0.470 | 0.292 | 0.596 | 0.494 |
| | Voyage-large-2 | 0.419 | 0.763 | 0.508 | 0.430 | 0.845 | 0.675 | 0.409 | 0.791 | 0.624 | 0.406 | 0.733 | 0.533 | 0.353 | 0.715 | 0.590 |
| | Fusion (4 models) | **0.519** | **0.881** | **0.655** | **0.601** | **0.876** | **0.825** | **0.566** | 0.888 | 0.818 | **0.544** | **0.881** | **0.756** | **0.476** | **0.814** | **0.719** |

Table 2: Pooling results by retrieval baselines (including fusion) in inference or oracle settings during FreshStack dataset construction. $\alpha$-N@10 denotes $\alpha$-nDCG@10, C@20 denotes Coverage@20 and R@50 denotes Recall@50. Stack Overflow Answer & Nuggets both rely on the gold answer for retrieval (oracle setting), whereas other methods do not rely on the gold answer for retrieval (inference setting). Overall, we highlight the best result in **bold** for each setting.

# 5 Pooling & Qualitative Evaluation

In this section, we attempt to answer RQ2 by evaluating methods that retrieve documents contributing to the judgment pools. We first evaluate the retrieval baselines during nugget-level support judgment (or sampling pools) with both inference and oracle settings.

In FreshStack, we are constructing a *test evaluation dataset*. Therefore, we can use the Stack Overflow answer or its variants in constructing judgment pools, as discussed previously in Section 3.4. We pool and sample documents from different systems and techniques, similar to how existing question answering datasets are constructed, such as Natural Questions [36] or XOR-TyDI [3], which assessed the document-level relevance by calculating the answer overlap in the document.

**Experimental Settings.** We perform retrieval with four techniques and baselines (as explained in Section 3.4) and an ensemble fusion of the top 100 documents, with each model score normalized and summed up. Evaluation metrics include $\alpha$-nDCG@10, Coverage@20, and Recall@50. We use GPT-4o with a temperature setting of 0.1[10] for both the automatic stages. Nugget generation uses a grading notes prompt with the question and answer, and support assessment uses a chain-of-thought prompt [83], judging up to a maximum of 20 documents simultaneously with a list of nuggets generated for each question. Finally, we sample and judge the top 20 fusion documents from each technique and setting (including the question) to avoid sampling holes, highlighting the importance of document diversity in our judgment pools.

## 5.1 Document Judgment Pooling Results

We outline the results achieved on document judgment pools during FreshStack construction with techniques from both inference and oracle settings. Key takeaways and findings are discussed below:

**Overall Highlights.** Table 2 reveals two key findings: (1) Techniques in the oracle setting significantly outperform techniques from the inference setting. We observe that both the Stack Overflow answer and nuggets techniques help pool documents relevant to the question, and (2) fusion outperforms all individual models, highlighting the value of diversity in model choice, aiding in the construction of our judgment pools in niche domains.

Within the inference setting in Table 2, we observe GPT-4o Sub-Questions achieves the best pooling results for four domains (except Yolo v7 & v8), showing that decomposing the question into smaller

---

[10]Separately, we tested temperatures of 0.1 and 0.7, observing an identical downstream retrieval accuracy during FreshStack construction.

sub-questions is useful in retrieving relevant documents. Stack Overflow Nuggets achieve the best results (except Laravel 10 & 11) in the oracle setting, showing that breaking down the answer into facts or nuggets is crucial. Amongst the individual models, BM25 achieves the best $\alpha$-nDCG@10 on all domains, asserting the importance of lexical approaches in judgment pool construction.

## 5.2 Qualitative Analysis

In our work, a crucial component is the automatic construction of nuggets and nugget-level document judgments with GPT-4o. To assess GPT-4o's accuracy, we calibrate with an expert human assessor (ML researcher) on a subset of LangChain, evaluating the quality of generated nuggets and nugget-document support labels for 60 randomly sampled questions.

### 5.2.1 Nugget Quality Evaluation

For nugget quality evaluation, we ask the human assessor to answer the following questions ($A$, $B$, and $C$) after reading the Stack Overflow question, answer, and list of nuggets. (1) $A$: Does the nugget produce hallucinated content? requiring a boolean response (2) $B$: Is the information provided in the nugget minor or redundant? also requiring a boolean response. After finishing $A$ and $B$, we ask (3) $C$: How many additional nuggets are required to cover all key ideas, requiring an integer in the response.

| Nugget Quality | | Judgment Quality | |
|---|---|---|---|
| Precision | 90.1 % | Relevant | 71.7 % |
| Recall | 96.6 % | Partially Relevant | 11.7 % |
| Groundedness | 96.4 % | Non-Relevant | 16.6 % |

Table 3: Expert evaluation of GPT-4o nugget quality and nugget-document relevance judgments on 60 sampled queries in LangChain.

**Evaluation Metrics.** We measure the nugget quality by calculating three metrics, by evaluating: (1) precision (P): whether nuggets generated are accurate, (2) recall (R): whether nuggets cover the key aspects of the answer and (3) groundedness (G): whether nuggets produce non-hallucinated content, i.e., within the scope of the answer. More formally, we define them as follows:

$$\mathbf{P} = \frac{|Nuggets| - \text{sum}(B)}{|Nuggets|}, \mathbf{R} = \frac{|Nuggets| - \text{sum}(B)}{|Nuggets| - \text{sum}(B) + C}, \mathbf{G} = \frac{|Nuggets| - \text{sum}(A)}{|Nuggets|} \quad (1)$$

where $|Nuggets|$ denotes the count of nuggets for a given question.

**Experimental Results.** As shown in Table 3, generated nuggets achieve above 90% in precision and 96% in recall and groundedness, indicating GPT-4o can generate high-quality nuggets required in the FreshStack framework. Most nuggets are well-grounded, i.e., do not produce hallucinated content (3.6% error), and cover the key aspects of the answer in terms of recall (3.4% error). Precision errors are higher (9.9% error), showing nuggets may contain either minor or repeated information. Within these errors, the last positioned nugget is not informative in almost 50% of all error cases, and either the first or second positioned nugget in the rest of the error cases.

### 5.2.2 Relevance Judgment Quality Evaluation

We assess the relevance between nuggets and documents in nugget-level support. Since judging all documents (including negatives) for each nugget is cumbersome, we qualitatively check for precision by evaluating only the relevant pairs. The annotator labels one positive document per question on a three-level scale: relevant, partially relevant, or non-relevant.

**Experimental Results.** As shown in Table 3, 71.7% of the judged nuggets and documents are relevant, including an additional 11.7% which are labeled partially relevant, indicating a high precision in GPT-4o support judgment. On the other hand, GPT-4o makes a mistake in judgment for 16.6% of the total questions. This discrepancy arises from several factors: some documents are relevant to only part of the nugget's information, leading to mislabeling; ambiguity within the nugget content can cause misjudgments; and occasionally, literal grounding of a document in the nugget does not translate to semantic relevance in answering the question.

## 6 Main Experiments

In this section, we evaluate retrievers and rerankers on document retrieval and RAG settings on the constructed FreshStack datasets, addressing RQ3 posed in our introduction. All models are evaluated

| Model | LangChain | | | Yolo v7 & v8 | | | Laravel 10 & 11 | | | Angular 16, 17 & 18 | | | Godot4 | | |
|---|---|---|---|---|---|---|---|---|---|---|---|---|---|---|---|
| | αN@10 | C@20 | R@50 | αN@10 | C@20 | R@50 | αN@10 | C@20 | R@50 | αN@10 | C@20 | R@50 | αN@10 | C@20 | R@50 |
| *Inference Setting: Retrieving documents using only the Stack Overflow (SO) query.* | | | | | | | | | | | | | | | |
| BM25 | 0.230 | 0.475 | 0.261 | 0.137 | 0.342 | 0.337 | 0.319 | 0.602 | 0.441 | 0.259 | 0.551 | 0.340 | 0.144 | 0.268 | 0.200 |
| BM25 + Reranker | 0.322 | 0.587 | 0.294 | 0.337 | 0.590 | 0.424 | 0.414 | 0.729 | 0.509 | 0.346 | 0.647 | 0.385 | 0.251 | 0.407 | 0.244 |
| BGE (Gemma-2) | 0.216 | 0.548 | 0.337 | 0.258 | 0.547 | 0.430 | 0.348 | 0.699 | 0.574 | 0.323 | 0.571 | 0.378 | 0.199 | 0.479 | 0.419 |
| BGE (Gemma-2) + Reranker | 0.349 | 0.662 | 0.387 | 0.388 | 0.666 | 0.459 | 0.306 | 0.646 | 0.571 | 0.296 | 0.595 | 0.387 | 0.324 | 0.576 | 0.471 |
| E5 Mistral (7B) | 0.304 | 0.654 | 0.393 | 0.243 | 0.552 | 0.394 | 0.250 | 0.565 | 0.470 | 0.262 | 0.548 | 0.368 | 0.217 | 0.444 | 0.359 |
| E5 Mistral (7B) + Reranker | 0.385 | 0.701 | 0.439 | 0.364 | 0.628 | 0.468 | 0.305 | 0.613 | 0.510 | 0.306 | 0.601 | 0.375 | 0.315 | 0.566 | 0.426 |
| Voyage-large-2 | 0.246 | 0.528 | 0.309 | 0.270 | 0.570 | 0.453 | 0.345 | 0.701 | 0.543 | 0.304 | 0.625 | 0.427 | 0.282 | 0.522 | 0.458 |
| Voyage-large-2 + Reranker | 0.345 | 0.648 | 0.355 | **0.418** | 0.670 | 0.514 | 0.302 | 0.653 | 0.529 | 0.300 | 0.600 | 0.414 | **0.342** | 0.598 | 0.511 |
| Fusion (4 models) | 0.337 | 0.700 | 0.477 | 0.304 | 0.627 | 0.534 | **0.426** | **0.748** | **0.646** | **0.385** | **0.719** | **0.532** | 0.265 | 0.550 | 0.505 |
| Fusion (4 models) + Reranker | **0.397** | **0.729** | **0.501** | 0.416 | **0.733** | **0.592** | 0.319 | 0.671 | 0.614 | 0.318 | 0.641 | 0.488 | 0.340 | **0.627** | **0.545** |
| *Best Scores in the Oracle Setting taken from Table 2: Upper Baselines on the FreshStack dataset* | | | | | | | | | | | | | | | |
| SO Answer: Fusion (4 models) | 0.484 | 0.821 | 0.619 | 0.546 | 0.854 | 0.788 | 0.564 | 0.892 | 0.820 | 0.470 | 0.805 | 0.695 | 0.449 | 0.741 | 0.683 |
| SO Nuggets: Fusion (4 models) | 0.519 | 0.881 | 0.655 | 0.601 | 0.876 | 0.825 | 0.566 | 0.888 | 0.818 | 0.544 | 0.881 | 0.756 | 0.476 | 0.814 | 0.719 |

Table 4: Document retrieval results on FreshStack with retrieval and reranker baselines (including fusion). Best scores or upper baselines in the oracle setting are taken from Table 2. The reranker is the Voyage AI rerank-2 model [1] reranking the top 100 documents. If the reranker improves upon the retrieval model, it is highlighted in green else red. We highlight the best result in **bold**.

using only the Stack Overflow question to retrieve documents in the inference setting, and do not include any information about the Stack Overflow answer or nuggets, ensuring a fair assessment.

**Experimental Settings.** We evaluate the same retrieval models used as baselines during pooling in FreshStack: BM25, BGE (Gemma-2), E5 Mistral 7B, Voyage-large-2, and Fusion. In addition, we evaluate the Voyage AI rerank-2 [1] as the reranker with a 16K context length, reranking the top 100 documents retrieved from each first-stage retrieval system and fusion. Metrics used for evaluation are defined in Section 4.3: $\alpha$-nDCG@10, Coverage@20, and Recall@50.

For RAG evaluation, we generate a RAG answer naively with five LLM generators: GPT-4o-mini, GPT-4o, GPT-4.1 (nano, mini), and GPT-4.1. We feed the query and the top 20 retrieved documents concatenated together as context. Next, we evaluate whether the RAG answer supports each nugget generated in Section 3.3, following Pradeep et al. [57], providing three labels: support, partial_support, or no_support. We compute the All Strict ($A_{strict}$) metric for RAG evaluation.

## 6.1 Document Retrieval Results

**Accuracy gap between oracle indicates plenty of headroom.** From Table 4, we observe techniques from the oracle setting (using Stack Overflow answers or nuggets) achieve a substantially higher $\alpha$-nDCG@10, Coverage@20, and Recall@50 in contrast to all models, including ensemble fusion and reranking with VoyageAI rerank-2. This highlights the complexity of answering FreshStack questions and demonstrates the headroom for improvement in existing code-mixed retrieval models to decrease the gap between retrieval models at inference and oracle approaches.

**Ensemble fusion outperforms individual models.** Individual retrieval models demonstrate limited success on the FreshStack dataset; whereas, the ensemble fusion of four retrieval models outperforms each retrieval model across all metrics ($\alpha$-nDCG@10, Coverage@20, and Recall@50) and all five domains, except $\alpha$-nDCG@10 on Godot4. This highlights a crucial point: a compound retrieval system [89], developed as an ensemble of retrieval models or something similar, is required to retrieve documents for niche and challenging domains, at present. However, fusion is inefficient at inference time, as it adds up individual model inferences, requiring alternatives.

**Opportunities to improve reranking.** When using a weak first-stage retrieval, neural rerankers typically improve document ranking [74], although it has been recently shown that this is not always the case when a strong first-stage retrieval is used [90, 26]. Consistent with these recent observations, reranking provides benefits over BM25 for all domains in the FreshStack dataset. However, for our dense retrievers, reranking provides a clear benefit on some but not all datasets. Specifically, while the reranker enhances $\alpha$-nDCG@10, Coverage@20, and Recall@50 for LangChain, Yolo v7 & v8, and Godot4, it reduces those metrics on Laravel 10 & 11 and Angular 16, 17 & 18 for BGE (Gemma-2), Voyage-large-2 and fusion. We suspect the reranker is better in certain programming languages such as Python, and we keep it as future work to understand the limitations of the neural reranker [26].

| Technique | Retrieval | Generator | LangChain | | | Yolo v7 & v8 | | | Laravel 10 & 11 | | | Angular 16, 17 & 18 | | | Godot4 | | |
|---|---|---|---|---|---|---|---|---|---|---|---|---|---|---|---|---|---|
| | | | nano | mini | full | nano | mini | full | nano | mini | full | nano | mini | full | nano | mini | full |
| *Inference Setting: Retrieving documents using only the Stack Overflow query.* | | | | | | | | | | | | | | | | | |
| Closed | No Retrieval | GPT-4o | – | 0.395 | 0.524 | – | 0.461 | 0.591 | – | 0.512 | 0.574 | – | 0.486 | 0.568 | – | 0.415 | 0.518 |
| Book | No Retrieval | GPT-4.1 | 0.444 | 0.517 | 0.564 | 0.470 | 0.663 | 0.647 | 0.557 | 0.646 | 0.621 | 0.508 | 0.604 | 0.597 | 0.483 | 0.616 | 0.573 |
| | Fusion | GPT-4o | – | 0.464 | 0.568 | – | 0.477 | 0.630 | – | 0.557 | 0.635 | – | 0.536 | 0.629 | – | 0.452 | 0.544 |
| StackOverflow | Fusion | GPT-4.1 | 0.438 | 0.578 | 0.610 | 0.571 | 0.649 | 0.624 | 0.572 | 0.668 | 0.660 | 0.575 | 0.670 | 0.674 | **0.492** | 0.573 | 0.595 |
| Query | Fusion + Rerank | GPT-4o | – | 0.444 | 0.587 | – | 0.492 | 0.625 | – | 0.545 | 0.617 | – | 0.551 | 0.620 | – | 0.428 | 0.551 |
| | Fusion + Rerank | GPT-4.1 | 0.467 | 0.594 | 0.625 | 0.527 | **0.679** | **0.684** | 0.583 | 0.657 | 0.651 | 0.564 | 0.663 | 0.650 | 0.491 | 0.578 | 0.591 |
| *Oracle Setting (Upper Baseline): Using the Stack Overflow answer directly or its variants for retrieval of documents within the corpus* | | | | | | | | | | | | | | | | | |
| StackOverflow | Fusion | GPT-4o | – | 0.492 | 0.618 | – | 0.559 | 0.668 | – | 0.549 | 0.656 | – | 0.584 | 0.680 | – | 0.477 | 0.576 |
| Nuggets | Fusion | GPT-4.1 | **0.533** | **0.654** | **0.651** | **0.591** | 0.667 | 0.667 | **0.607** | **0.681** | **0.696** | **0.626** | **0.717** | **0.709** | 0.489 | **0.628** | **0.668** |

Table 5: RAG evaluation results measuring nugget recall with All Strict ($A_{strict}$) scores on LLM-generated answer with: GPT-4o-mini and GPT-4o, GPT-4.1 (nano, mini) and GPT-4.1. The knowledge cutoff date for GPT-4o series is October 2023 and GPT-4.1 series is June 2024.

## 6.2 RAG Evaluation Results

**Retrieved context is key for RAG accuracy.** From the evaluation results measuring $A_{strict}$ capturing nugget recall in Table 5, the RAG answer generated in the oracle setting outperforms other techniques except Yolo v7 & v8, indicating that the oracle retrieved context is the key. Next, we evaluate state-of-the-art generators (e.g., GPT-4.1) for evaluation to establish stronger baselines and oracle results, and to check for possible dataset contamination. Consistent across all observations, GPT-4.1 performs better than GPT-4o across all domains for the closed-book setting, which we suspect is due to a more recent knowledge cutoff date (June 2024 versus October 2023), covering all questions asked in FreshStack. Lastly, the fusion + rerank inference setting with GPT-4.1 as the generator is competitive, outperforming even the oracle fusion setting in Yolo v7 & v8.

**FreshStack is *not* saturated.** The current numbers achieved on FreshStack should not be taken as evidence that the benchmark is "saturated". Rather, they reflect a meaningful starting point, incorporating complex queries drawn from niche and recent domains. This design presents an ongoing challenge for RAG systems' generalization, shared by other benchmarks such as BRIGHT [71]. Moreover, FreshStack has been recognized by its inclusion in the RTEB benchmark [16].

## 7 Discussion & Future Work

**Generalization Assumption**. While FreshStack can be generally applied to any domain, we applied it in our work covering questions on technical documents. The key assumptions for FreshStack to generalize to a new domain are: (1) requires at least 50 queries (50 is the typical minimum number of queries used in TREC, e.g., Deep Learning (DL) track 2020 used 54 queries [12]). that are human-generated for a lower variance in retrieval performance, (2) answers to the queries (preferably human-generated) that can be linked using information available in documents, (3) a collection of at least 5–10K documents (or chunks) with rich information.

**Data Contamination.** Fully eliminating data contamination is infeasible, as both Stack Overflow and GitHub are publicly available. However, by utilizing temporally recent domains with niche domains and carefully considering knowledge cutoff dates in LLMs, we can potentially lower the likelihood of data contamination during LLM pre-training and supervised fine-tuning.

## 8 Conclusion

The emergence of RAG has improved modern retrieval systems by allowing real-time data incorporation into LLMs. However, existing IR and RAG benchmarks that measure retrieval quality are outdated. In this work, we introduce a holistic framework, FreshStack, to construct challenging datasets to evaluate retrieval systems realistically. We source real user questions and answers from Stack Overflow and build a document corpus using technical documents from public GitHub repositories. Using FreshStack, we construct datasets on five niche domains and evaluate four frontier retrieval models and a reranker model in the document retrieval setting. The accuracy gap observed between the retrieval models and approaches in the oracle setting indicates plenty of headroom for improvement, and we identify cases that may motivate future research in reranking. We hope FreshStack will encourage the community to build realistic IR and RAG benchmarks in the future.

## Acknowledgments and Disclosure of Funding

We thank Sean Kulinski and Alexander Trott for helping us set up the grading notes prompt required in nugget generation. We also thank Jacob Portes, Max Marion, Matei Zaharia, and others from Databricks who provided feedback at the early stages of the project.

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

# Technical Appendices and Supplementary Material

## A    Comparison of FreshStack with Existing IR and RAG Benchmarks

Table 6 compares FreshStack against existing code-focused IR or RAG benchmarks. Below, we briefly describe a few advantages of Fresh-Stack over existing RAG benchmarks:

First, the FreshStack framework utilizes user-asked questions and curated answers, making the evaluation challenging. A majority of existing benchmarks are unrealistic, derived from easily retrievable domains and queries, such as Neural Code Search [38], making it easy to retrieve and answer them, rather than being grounded in solving real user problems provided

| IR / RAG Benchmarks | Niche Domains | Complex Questions | Dynamic Updates | Challenge Level |
|---|---|---|---|---|
| CQADupstack [23] | No | No | No | Easy |
| CodeSearchNet [24] | No | No | No | Easy |
| COIR [39] | Limited | Yes | No | Moderate |
| Stack Overflow-QA [39] | No | Yes | No | Moderate |
| CodeRAG-Bench [82] | Limited | No | No | Moderate |
| Neural Code Search [38] | No | No | No | Moderate |
| SWE-Bench [29] | No | Yes | Yes | High |
| **FreshStack (ours)** | **Yes** | **Yes** | **Yes** | **High** |

Table 6: A comparison of existing IR/RAG evaluation benchmarks with FreshStack.

in FreshStack. We are not crafting artificial (or LLM-generated) questions or sampling questions myopically. Second, all answers in FreshStack are supported in real time by information from technical documentation in GitHub repositories. Third, the framework is designed to be general and scalable without modification. Finally, FreshStack is focused on niche domains and recent domains, taking careful measures to mitigate risks with data contamination introduced by LLMs, ensuring that the benchmark is not susceptible to distortion or leaderboard overfitting [68].

## B    Retrieval and RAG Evaluation Metrics

### B.1    Retrieval Evaluation Metrics

$\alpha$**-nDCG@k.** Introduced by Clarke et al. [9], this variant of Normalized Discounted Cumulative Gain (nDCG) measures search diversification. The $\alpha$ parameter is a geometric penalization for redundant documents, i.e., each redundant document achieves a penalization of $\times (1 - \alpha)$. Despite the metric being used for different user intents, we utilize it to ensure document rankings reference diverse nuggets in the answer. We would ask the reader to refer to Clarke et al. [9] for more information.

**Coverage@k.** The metric introduced in our work measures the average proportion of the nuggets covered by the top-k retrieved documents. The mathematical formula is calculated as:

$$\text{Coverage@k} = \frac{1}{|Q|} \sum_{q=1}^{Q} \frac{\left| \bigcup_{i=1}^{k} \text{Nuggets}(d_{qi}) \right|}{|\text{Nuggets}(q)|} \qquad (2)$$

where $Q$ contains all questions, $\text{Nuggets}(d_{qi})$ are nuggets supported by document $d_{qi}$ and $\text{Nuggets}(q)$ are nuggets for question $q$.

**Recall@k.** The standard relevance metric measures the proportion of relevant documents retrieved within the top-k results, out of all relevant documents for a given question. A document is judged relevant if it supports at least one nugget.

### B.2    RAG Evaluation Metric

**All Strict** ($A_{strict}$)    Introduced in Pradeep et al. [57], for each query, we have a list of nuggets generated from Section 3.3, and for each RAG answer generated, we have a record of which nuggets it contains, in terms of a three-way judgment: `support`, `partial_support`, and `no_support`. The final step is to compute the score for the RAG answer to a query $q$. The score of a run is simply the mean of the score for each query. We compute the following scores per query:

We calculate a score based on all nuggets in the RAG answer, but with strict nugget matching. For a given nugget $i$:

$$p_i = \begin{cases} 1 & \text{if assignment} = \texttt{support} \\ 0 & \text{otherwise} \end{cases} \qquad (3)$$

The "All Strict" score is then calculated as:

$$A_{strict} = \frac{\sum_i p_i}{|Nuggets|},$$

where $|Nuggets|$ denotes the count of nuggets for a given query $q$.

## C    FreshStack Instance Description

Each FreshStack dataset instance contains the following four components, as shown in Figure 1. A complete example of a dataset instance is shown in Table 9.

- **Question & Answer**: The title and body (description) of the Stack Overflow post as the question, with the accepted answer. The title is a short sentence, and the body contains the detailed issue with code snippets and/or outputs.
- **Nuggets**: The list of atomic facts highlighting the essential information in the Stack Overflow question and answer.
- **Document Corpus**: The exhaustive list of chunked source documents (code snippets, text documentation, etc.) compiled from GitHub repositories.
- **Relevance Judgments**: Unlike traditional IR benchmarks, such as BEIR [74], which contain question and document-level relevance judgments, FreshStack datasets contain nugget-level relevance judgments for document chunks.

## D    Discussion and Future Work

FreshStack is a holistic framework for building challenging IR and RAG evaluation datasets. We apply the framework to community-sourced questions (with curated answers) sourced from Stack Overflow and documents sourced from GitHub repositories. The framework is adaptable to other domains like Stack Exchange or internal forums.

**Maintaining FreshStack leaderboard.** We are actively maintaining a retrieval leaderboard by evaluating increasingly recent IR models on the document retrieval task for FreshStack. We have evaluated and included the following families of models: (i) Qwen3 embeddings (596M, 4B, and 8B) [93], (ii) Jina embeddings (V3 and V4) [70, 21], (iii) Stella (1.5B and 400M) [91], (iv) OpenAI text-embedding-3 (small and large) [53], (v) GTE-large-en-v1.5 [41], (vi) Nomic Embed (Code) [72], and (vii) CodeRankEmbed [72]. The updated results are provided in Table 7. Recent dense retrieval models, such as Qwen3-8B (embedding), continue to improve and perform competitively on Fresh-Stack, even outperforming the strong fusion baseline on two domains: Yolo v7 & v8 and Godot 4. We will continue to benchmark newer models as they are released.[11]

**Extending RAG Evaluation.** We focused on the evaluation of the retrieval setting primarily due to two reasons: (1) existing RAG datasets evaluate retrieval using relevance criteria only, however, we evaluated models based on both diversity and relevance criteria, and (2) a crucial step in FreshStack is sourcing and building a document corpus and developing a general framework for high-quality pools and automatic judgments, which we can evaluate better in the retrieval setting. We evaluated the quality of LLM-based answer generation with nugget-based recall with $A_{strict}$ metric [56, 57], which calculates how many nuggets are supported within a system's response. However, we keep an in-depth evaluation of the RAG answer, accounting for factors such as support [73], as future work.

**Benchmark Contamination.** The FreshStack dataset is built on Stack Overflow data, making it susceptible to future data contamination. Newer released LLMs such as GPT-4.1 with a recent knowledge cutoff date[12] of June 2024, provide the possibility of dataset contamination with GPT-4.1 as FreshStack queries originated from January 2023 until June 2024 (as shown in Figure 3). To mitigate this potential data contamination, the FreshStack framework can add newer asked user questions in existing domains, retire old and potentially contaminated domains, and add newer domains that form in the future. The relevance of FreshStack in the community relies on a continued commitment to keeping it updated in the upcoming years.

---

[11]The updated FreshStack leaderboard: https://fresh-stack.github.io/#leaderboard.
[12]https://help.openai.com/en/articles/9624314-model-release-notes

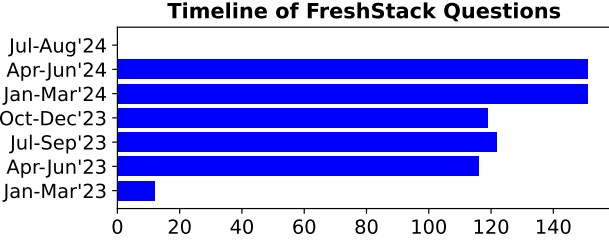

Figure 3: Timeline versus frequency of how many FreshStack queries were asked on Stack Overflow in every quarter. All queries included in FreshStack were asked between January 2023 and June 2024, with the highest frequencies observed in 2024, showing the growing importance of all five domains.

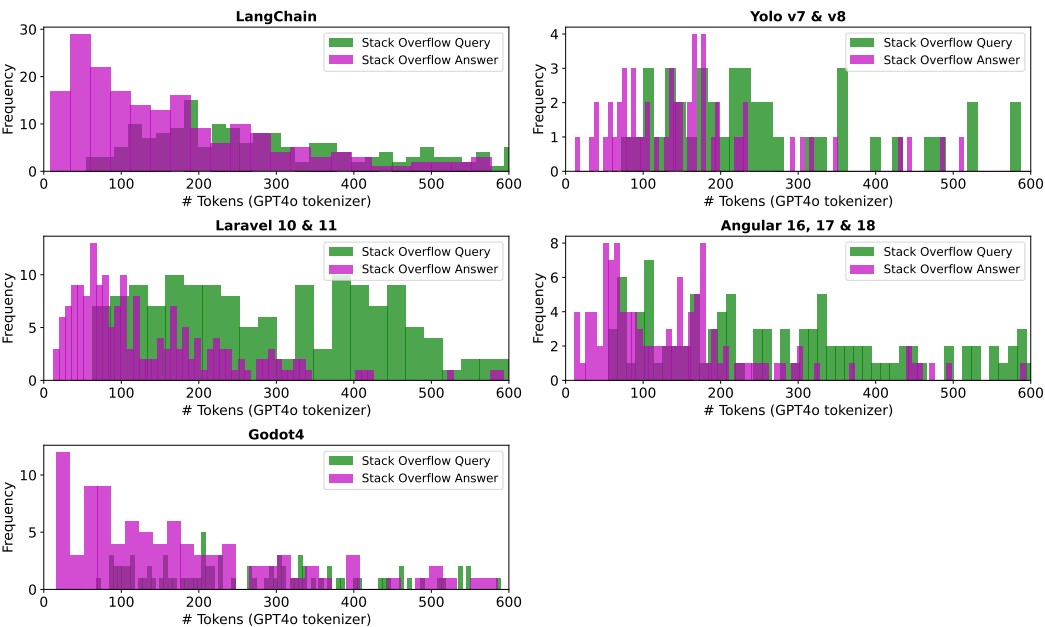

Figure 4: Token distribution (using the GPT-4o tokenizer) of Stack Overflow questions and answers for all domains in FreshStack. Unlike other benchmarks, questions in FreshStack (highlighted in green) are much longer than answers in FreshStack (highlighted in maroon).

| Model | LangChain | | | Yolo v7 & v8 | | | Laravel 10 & 11 | | | Angular 16, 17 & 18 | | | Godot4 | | |
|---|---|---|---|---|---|---|---|---|---|---|---|---|---|---|---|
| | $\alpha$N@10 | C@20 | R@50 | $\alpha$N@10 | C@20 | R@50 | $\alpha$N@10 | C@20 | R@50 | $\alpha$N@10 | C@20 | R@50 | $\alpha$N@10 | C@20 | R@50 |
| *Inference Setting: Retrieving documents using only the Stack Overflow (SO) query.* | | | | | | | | | | | | | | | |
| Fusion (4 models) | **0.337** | **0.700** | **0.477** | 0.304 | 0.627 | 0.534 | **0.425** | **0.748** | **0.646** | **0.385** | **0.719** | **0.532** | 0.265 | 0.550 | 0.505 |
| Qwen3-8B (embedding) [93] | 0.331 | 0.694 | 0.423 | 0.393 | 0.728 | **0.567** | 0.421 | **0.748** | 0.615 | 0.373 | 0.700 | 0.502 | **0.307** | **0.576** | **0.521** |
| Qwen3-4B (embedding) [93] | 0.320 | 0.675 | 0.415 | **0.404** | **0.744** | 0.550 | 0.402 | **0.748** | 0.604 | 0.304 | 0.618 | 0.442 | 0.303 | 0.496 | 0.440 |
| Stella-1.5B [91] | 0.315 | 0.660 | 0.388 | 0.334 | 0.624 | 0.559 | 0.370 | 0.681 | 0.590 | 0.330 | 0.630 | 0.414 | 0.237 | 0.481 | 0.443 |
| Voyage-large-2 | 0.246 | 0.528 | 0.308 | 0.270 | 0.570 | 0.453 | 0.345 | 0.701 | 0.543 | 0.304 | 0.625 | 0.427 | 0.282 | 0.522 | 0.458 |
| Jina V4 (embedding) [21] | 0.277 | 0.596 | 0.379 | 0.311 | 0.692 | 0.524 | 0.324 | 0.677 | 0.552 | 0.279 | 0.539 | 0.321 | 0.220 | 0.416 | 0.351 |
| Stella-400M [91] | 0.285 | 0.608 | 0.356 | 0.241 | 0.538 | 0.447 | 0.320 | 0.648 | 0.534 | 0.288 | 0.619 | 0.359 | 0.244 | 0.476 | 0.412 |
| BGE (Gemma-2) | 0.216 | 0.548 | 0.337 | 0.258 | 0.547 | 0.430 | 0.348 | 0.699 | 0.574 | 0.323 | 0.571 | 0.378 | 0.200 | 0.479 | 0.419 |
| Qwen3-0.6B (embedding) [93] | 0.259 | 0.588 | 0.369 | 0.260 | 0.504 | 0.383 | 0.288 | 0.593 | 0.463 | 0.253 | 0.535 | 0.356 | 0.249 | 0.495 | 0.400 |
| E5 Mistral (7B) | 0.304 | 0.654 | 0.393 | 0.243 | 0.552 | 0.394 | 0.250 | 0.565 | 0.470 | 0.262 | 0.548 | 0.368 | 0.217 | 0.444 | 0.359 |
| text-embedding-3-large [53] | 0.207 | 0.507 | 0.292 | 0.275 | 0.585 | 0.412 | 0.298 | 0.627 | 0.494 | 0.271 | 0.556 | 0.353 | 0.187 | 0.409 | 0.316 |
| Jina V3 (embedding) [70] | 0.223 | 0.533 | 0.299 | 0.188 | 0.448 | 0.338 | 0.309 | 0.654 | 0.489 | 0.224 | 0.536 | 0.293 | 0.190 | 0.405 | 0.301 |
| GTE (large) v1.5 [41] | 0.206 | 0.470 | 0.252 | 0.195 | 0.445 | 0.271 | 0.318 | 0.626 | 0.482 | 0.284 | 0.578 | 0.343 | 0.127 | 0.348 | 0.240 |
| BM25 | 0.230 | 0.475 | 0.261 | 0.137 | 0.342 | 0.337 | 0.319 | 0.602 | 0.441 | 0.259 | 0.551 | 0.340 | 0.144 | 0.268 | 0.200 |
| Nomic Embed (code) [72] | 0.224 | 0.518 | 0.292 | 0.227 | 0.539 | 0.390 | 0.222 | 0.532 | 0.407 | 0.237 | 0.511 | 0.356 | 0.178 | 0.341 | 0.295 |
| text-embedding-3-small [53] | 0.213 | 0.523 | 0.283 | 0.172 | 0.423 | 0.303 | 0.245 | 0.571 | 0.438 | 0.214 | 0.491 | 0.295 | 0.197 | 0.392 | 0.330 |
| CodeRankEmbed [72] | 0.099 | 0.271 | 0.128 | 0.075 | 0.215 | 0.128 | 0.108 | 0.324 | 0.225 | 0.146 | 0.363 | 0.167 | 0.091 | 0.224 | 0.160 |

Table 7: Document retrieval results on FreshStack under the *inference setting*, using only the Stack Overflow (SO) query. Metrics include $\alpha$-N@10 ($\alpha$-nDCG@10), C@20 (Coverage@20) and R@50 (Recall@50). Updated results (including average scores across five domains) are available at the following website: https://fresh-stack.github.io/#leaderboard.

| Domain | GitHub Repository | License |
|---|---|---|
| **LangChain** | [1] https://github.com/langchain-ai/langchain | MIT |
| | [2] https://github.com/langchain-ai/langchainjs | MIT |
| | [3] https://github.com/langchain-ai/langchain-nextjs-template | MIT |
| | [4] https://github.com/chroma-core/chroma | Apache-2.0 |
| | [5] https://github.com/openai/openai-cookbook | MIT |
| | [6] https://github.com/openai/openai-python | Apache-2.0 |
| | [7] https://github.com/run-llama/llama_index | MIT |
| | [8] https://github.com/Azure-Samples/openai | MIT |
| | [9] https://github.com/Azure-Samples/azure-search-openai-demo | MIT |
| | [10] https://github.com/huggingface/transformers | Apache-2.0 |
| **Yolo v7 & v8** | [1] https://github.com/ultralytics/ultralytics | AGPL-3.0 |
| | [2] https://github.com/ultralytics/docs | AGPL-3.0 |
| | [3] https://github.com/pytorch/pytorch | Modified BSD |
| | [4] https://github.com/WongKinYiu/yolov7 | GPL-3.0 |
| | [5] https://github.com/opencv/opencv | Apache-2.0 |
| **Laravel 10 & 11** | [1] https://github.com/laravel/framework | MIT |
| | [2] https://github.com/laravel/laravel | MIT |
| | [3] https://github.com/laravel/laravel.com | MIT |
| | [4] https://github.com/laravel/docs | MIT |
| | [5] https://github.com/laravel/breeze | MIT |
| | [6] https://github.com/livewire/livewire | MIT |
| | [7] https://github.com/php/php-src | PHP |
| | [8] https://github.com/php/doc-en | PHP |
| | [9] https://github.com/php/web-php | PHP |
| **Angular 16, 17 & 18** | [1] https://github.com/angular/angular | MIT |
| | [2] https://github.com/angular/components | MIT |
| | [3] https://github.com/angular/angular-cli | MIT |
| | [4] https://github.com/microsoft/TypeScript | Apache-2.0 |
| **Godot4** | [1] https://github.com/godotengine/godot | MIT |
| | [2] https://github.com/godotengine/godot-demo-projects | MIT |
| | [3] https://github.com/godotengine/godot-docs | CC BY 3.0 |
| | [4] https://github.com/godotengine/godot-website | MIT |
| | [5] https://github.com/GDQuest/learn-gdscript | MIT |
| | [6] https://github.com/dotnet/csharplang | GPL |

Table 8: GitHub repositories and their licenses used to construct the document collection for each domain in FreshStack. All repositories were downloaded and chunked on 22$^{nd}$ October 2024.

| LangChain | Query ID: 78256389 |
|---|---|

| **Stack Overflow Query** | **Title**: Chromadb from_documents function giving error. 
 **Text:** The following function was working till a few days ago but now gives this error: ValueError: Expected EmbeddingFunc- tion._call_ to have the following signature: odict_keys(['self', 'input']), got odict_keys(['args', 'kwargs']) Please see https:// docs.trychroma.com/embeddings for details of the 'EmbeddingFunction' interface. Please note the recent change to the 'Em- beddingFunction' interface: https://docs.trychroma.com/migration#migration-to-0416---november-7-2023 I am not sure what changes are necessary to work with this. |

```python
1   def create_chromadb(link):
2       embedding_function = SentenceTransformerEmbeddings(model_name="all-MiniLM-L6-v2")
3       loader = TextLoader(link)
4       documents = loader.load()
5
6       # Split the documents into chunks (no changes needed here)
7       text_splitter = RecursiveCharacterTextSplitter(chunk_size=1000, chunk_overlap=500)
8       chunks = text_splitter.split_documents(documents)
9
10      # Update for new EmbeddingFunction definition
11      # D is set to the type of documents (Text in this case)
12      D = Union[str, List[str]] # Adjust based on your document format (single string or list of strings)
13      embedding_function: EmbeddingFunction[D] = embedding_function
14
15      # Initialize Chroma with the embedding function and persist the database
16      db = Chroma.from_documents(chunks, embedding_function, ids=None, collection_name="langchain", ↘
                    persist_directory="./chroma_db")
17      db.persist()
18      print(f"Saved {len(chunks)} chunks")
19
20      return db
```

| **Stack Overflow Answer** | I slightly modify your code, using 'HuggingFaceEmbeddings' instead of 'SentenceTransformerEmbeddings'. |

```python
1   from langchain_community.embeddings import HuggingFaceEmbeddings
2   embedding = HuggingFaceEmbeddings(model_name="sentence-transformers/all-MiniLM-L6-v2")
3
4   from langchain_community.vectorstores import Chroma
5   db = Chroma.from_documents(documents=chunks, embedding=embedding, persist_directory="/tmp/chroma_db")
6   db.persist()
```

| **GPT-4o Nuggets** | 1. The error is due to a mismatch in the function signature expected by 'Chroma.from_documents' when using 'SentenceTrans- formerEmbeddings'. 
 2. Use 'HuggingFaceEmbeddings' instead of 'SentenceTransformerEmbeddings' to resolve the error. 
 3. Import 'HuggingFaceEmbeddings' from 'langchain_community.embeddings'. 
 4. Initialize 'HuggingFaceEmbeddings' with the model name "sentence-transformers/all-MiniLM-L6-v2". 
 5. Pass the initialized 'HuggingFaceEmbeddings' to the 'Chroma.from_documents' function. |

| **Retrieved GitHub Document** | **Document ID**: langchain/templates/intel-rag-xeon/ingest.py_0_1486 
 **Supported Nuggets:** *Nugget 2*, *Nugget 4*, and *Nugget 5*. |

```python
1   import os
2
3   from langchain.text_splitter import RecursiveCharacterTextSplitter
4   from langchain_chroma import Chroma
5   from langchain_community.document_loaders import UnstructuredFileLoader
6   from langchain_community.embeddings import HuggingFaceEmbeddings
7   from langchain_core.documents import Document
8
9   def ingest_documents():
10      """
11      Ingest PDF to Redis from the data/ directory that
12      contains Edgar 10k filings data for Nike.
13      """
14      # Load list of pdfs
15      data_path = "data/"
16      doc = [os.path.join(data_path, file) for file in os.listdir(data_path)][0]
17
18      print("Parsing 10k filing doc for NIKE", doc)
19      text_splitter = RecursiveCharacterTextSplitter(chunk_size=1500, chunk_overlap=100, add_start_index=↘
                    True)
20      loader = UnstructuredFileLoader(doc, mode="single", strategy="fast")
21      chunks = loader.load_and_split(text_splitter)
22
23      print("Done preprocessing. Created", len(chunks), "chunks of the original pdf")
24
25      # Create vectorstore
26      embedder = HuggingFaceEmbeddings(model_name="sentence-transformers/all-MiniLM-L6-v2")
27
28      documents = []
29      for chunk in chunks:
30          doc = Document(page_content=chunk.page_content, metadata=chunk.metadata)
31          documents.append(doc)
32
33      # Add to vectorDB
34      _ = Chroma.from_documents(documents=documents, collection_name="xeon-rag", embedding=embedder, ↘
                    persist_directory="/tmp/xeon_rag_db")
35
36
37  if __name__ == "__main__":
38      ingest_documents()
```

Table 9: A complete example of a dataset instance from LangChain in FreshStack. The relevant nuggets supported by the retrieved GitHub document are highlighted in green.

