# OpenReview forum: "FreshStack: Building Realistic Benchmarks for Evaluating Retrieval on Technical Documents"
_NeurIPS.cc/2025/Datasets_and_Benchmarks_Track — NeurIPS 2025 Datasets and Benchmarks Track poster_

### Official Review · Reviewer_KD62 · 2025-07-02

**Ethics Flags:** Data privacy, copyright, and consent,…
**Rating:** 5
**Confidence:** 4

**Summary:**

The paper introduces FreshStack, a benchmark suite for evaluating information retrieval (IR) on technical documents.

The benchmark is constructed using real user queries and answers sourced from Stack Overflow, combined with technical documentation from GitHub.

It supports nugget-level relevance judgment via GPT-4, simulating user-intended facts.

The authors provide datasets across five technical domains, propose a structured framework for data construction, and benchmark several retriever models and RAG systems.

**Dataset Code Accessibility:**

Yes

**Dataset Code Comments:**

The authors link to their dataset and code at https://fresh-stack.github.io/, which appears well-documented and accessible.

**Ethical Comments:**

See Limitation parts.

**Ethical Considerations:**

Yes, there are ethics concerns that require attention by the authors

**Final Justification:**

Thanks to the authors for the thoughtful rebuttal. As said in the final review round. I am keep current positve scores of acceptance.

**Limitations Weaknesses:**

1) The benchmark currently only includes five domains (ML, CV, frontend, game development), which may limit generalizability.

2) Terms such as "inference" and "oracle" settings are used frequently but are not clearly defined until later in the paper. Earlier clarification or a table summarizing all settings would help.

3) Why Oracle as Gold Standard?: The paper uses oracle systems as a gold reference but does not fully justify their superiority over human-created baselines.

4) While some comparisons to related work are provided, the distinction between FreshStack and Neural Code Search (NCS) could be elaborated further, particularly regarding data construction

5) While qualitative evaluation is mentioned, more concrete examples on failure modes (e.g., hallucinated nuggets) would be helpful.

6) The dataset is constructed from publicly available resources (Stack Overflow and GitHub), and privacy or consent issues are not apparent. Those Data has already been used in GPT like model training, how you declare or test later those model performance and benchmarking them:)

**Strengths Contributions:**

1） Use of real Stack Overflow Q&A pairs ensures high relevance and realism. The focus on nugget-level support enables fine-grained and explainable evaluation.

2）The benchmark framework includes clearly defined steps (Figure 2), including community query sourcing, nugget generation, document chunking, pooling, and evaluation.

3）The paper benchmarks a variety of retrieval models (e.g., BGE, E5, BM25, fusion strategies), and compares both inference and oracle settings.

4）Well-written, logically structured, and includes informative tables and figures (e.g., Table 1, Table 2). Clearly Why & Why us Flow

---

> ### Author Rebuttal · Authors · 2025-07-31
>
> Thank you for your review for summarizing our paper’s strengths and mentioning insightful weaknesses. We appreciate that you like our use of Stack Overflow Q&A pairs, clearly defined steps in Figure 2, and our benchmarking a variety of models. For weaknesses in (2) and (4), we will clearly define the “inference” and “oracle” settings, elaborate on the distinction between FreshStack and Neural Code Search in the camera-ready version, and thank you for highlighting these weaknesses. Other concerns raised require more explanation, and we have addressed them in detail below:
>
> 1. **We focus on evaluating more recent IR models on FreshStack, which spans fewer domains but has far more queries than typical IR benchmarks**: Yes, we initially selected a few but diverse domains (check Table 1 for summary) to test out our benchmark construction approach in FreshStack. We’re particularly interested in whether we can design fresh benchmarks in the face of rapid progress in AI research. For this reason, instead of expanding the current domains, we have further invested in evaluating more and newer IR models on FreshStack. We have evaluated and added the following family of models: (i) Qwen-3 embeddings (596M, 4B & 8B), (ii) Jina-embeddings (V3 & V4), (iii) Stella V5 (1.5B & 400M), (iv) OpenAI text-embedding-3 (small & large), (v) GTE-large-en-v1.5, (vi) Nomic Embed (Code) and (vii) CodeRankEmbed on FreshStack. We will add these baselines to the camera-ready version.
>
> - Related IR benchmarks such as FollowIR [1] also include a small set of domains for evaluation (three domains: TREC-News, TREC-Core and Robust04). In contrast to typical TREC datasets which contain a few hundred queries (TREC-DL 2020 contains 54 queries [2], TREC Robust04 contains 249 queries [3], TREC RAG 2024 contains 301 queries [4]), FreshStack contains 672 questions spanning five domains, that is higher than typical IR evaluation datasets.
>
> | | LangChain | LangChain | LangChain | Yolo v7 & v8 | Yolo v7 & v8 | Yolo v7 & v8 | Laravel 10 & 11 | Laravel 10 & 11 | Laravel 10 & 11 | Angular 16, 17 & 18 | Angular 16, 17 & 18| Angular 16, 17 & 18| Godot4 | Godot4 | Godot4 |
> | ----------------------- | ----- | -----  | ------ | ----  | ------ | -----  | -----   | ---- | ---- | -----   | ---- | ---- | --- | -----     | -----   |
> | **Model Name**              | **αN@10** | **C@20**  | **R@50** | **αN@10** | **C@20**  | **R@50** | **αN@10** | **C@20**  | **R@50** | **αN@10** | **C@20**  | **R@50** | **αN@10** | **C@20**  | **R@50** |
>  |
> | Qwen3-8B (Emb)          | 0.331 | 0.694  | 0.423  | 0.393 | 0.728  | 0.567  | 0.421   | 0.748| 0.615| 0.373   | 0.700| 0.502| 0.307 | 0.576   | 0.521   |
> | Qwen3-4B (Emb)          | 0.320 | 0.675  | 0.415  | 0.404 | 0.744  | 0.550  | 0.402   | 0.748| 0.604| 0.304   | 0.618| 0.442| 0.303 | 0.496   | 0.440   |
> | Fusion (4 models)       | 0.337 | 0.700  | 0.477  | 0.304 | 0.627  | 0.534  | 0.425   | 0.748| 0.646| 0.385   | 0.719| 0.532| 0.265 | 0.550   | 0.505   |
> | Stella-1.5B v5          | 0.315 | 0.660  | 0.388  | 0.334 | 0.624  | 0.559  | 0.370   | 0.681| 0.590| 0.330   | 0.630| 0.414| 0.237 | 0.481   | 0.443   |
> | Voyage Large 2          | 0.246 | 0.528  | 0.308  | 0.270 | 0.570  | 0.453  | 0.345   | 0.701| 0.543| 0.304   | 0.625| 0.427| 0.282 | 0.522   | 0.458   |
> | Jina V4 (Emb)           | 0.277 | 0.596  | 0.379  | 0.311 | 0.692  | 0.524  | 0.324   | 0.677| 0.552| 0.279   | 0.539| 0.321| 0.220 | 0.416   | 0.351   |
> | Stella-400M v5          | 0.285 | 0.608  | 0.356  | 0.241 | 0.538  | 0.447  | 0.320   | 0.648| 0.534| 0.288   | 0.619| 0.359| 0.244 | 0.476   | 0.412   |
> | BGE (Gemma-2)           | 0.216 | 0.548  | 0.337  | 0.258 | 0.547  | 0.430  | 0.348   | 0.699| 0.574| 0.323   | 0.571| 0.378| 0.200 | 0.479   | 0.419   |
> | Qwen3-0.6B (Emb)        | 0.259 | 0.588  | 0.369  | 0.260 | 0.504  | 0.383  | 0.288   | 0.593| 0.463| 0.253   | 0.535| 0.356| 0.249 | 0.495   | 0.400   |
> | E5 (Mistral-7B)         | 0.304 | 0.654  | 0.393  | 0.243 | 0.552  | 0.394  | 0.250   | 0.565| 0.470| 0.262   | 0.548| 0.368| 0.217 | 0.444   | 0.359   |
> | text-embedding-3-large  | 0.207 | 0.507  | 0.292  | 0.275 | 0.585  | 0.412  | 0.298   | 0.627| 0.494| 0.271   | 0.556| 0.353| 0.187 | 0.409   | 0.316   |
> | Jina V3 (Emb)           | 0.223 | 0.533  | 0.299  | 0.188 | 0.448  | 0.338  | 0.309   | 0.654| 0.489| 0.224   | 0.536| 0.293| 0.190 | 0.405   | 0.301   |
> | GTE (large) v1.5        | 0.206 | 0.470  | 0.252  | 0.195 | 0.445  | 0.271  | 0.318   | 0.626| 0.482| 0.284   | 0.578| 0.343| 0.127 | 0.348   | 0.240   |
> | BM25                    | 0.230 | 0.475  | 0.261  | 0.137 | 0.342  | 0.337  | 0.319   | 0.602| 0.441| 0.259   | 0.551| 0.340| 0.144 | 0.268   | 0.200   |
> | Nomic Embed (Code)      | 0.224 | 0.518  | 0.292  | 0.227 | 0.539  | 0.390  | 0.222   | 0.532| 0.407| 0.237   | 0.511| 0.356| 0.178 | 0.341   | 0.295   |
> | text-embedding-3-small  | 0.213 | 0.523  | 0.283  | 0.172 | 0.423  | 0.303  | 0.245   | 0.571| 0.438| 0.214   | 0.491| 0.295| 0.197 | 0.392   | 0.330   |
> | CodeRankEmbed           | 0.099 | 0.271  | 0.128  | 0.075 | 0.215  | 0.128  | 0.108   | 0.324| 0.225| 0.146   | 0.363| 0.167| 0.091 | 0.224   | 0.160   |
>
> 2. **Why is Oracle the gold standard?** A standard IR evaluation dataset is a test collection as defined in the Cranfield paradigm, i.e., it contains three components: (i) a fixed set of queries, (ii) a corpus containing documents, and (iii) relevance judgments between queries and documents from the corpus. Therefore, it is not typical of an IR collection to have gold or human-curated *answers* for the queries. FreshStack, on the other hand, has gold answers for queries that make it possible to use them as Oracle, to sample better documents and easier to label documents that support the answer. For comparison, TREC-BioGen [5] track required annotators to label answers and documents.
>
> 3. **More concrete examples on failure modes**: Thank you for mentioning this. We have numbers already available from our manual analysis (Table 3). We will include more examples of failure methods: (a) hallucinated or non-relevant nuggets and (b) incorrect support level between nugget and document in the camera-ready version.
>
> 4. **Privacy and data contamination**: We agree that data contamination is possible and unavoidable, as both StackOverflow and GitHub are publicly available. However, with temporally recent topics and carefully considering knowledge cutoff dates in LLMs, we can potentially lower the possibility of data contamination. By releasing the code for the FreshStack pipeline, we hope to make this easier.
>
> 5. **Data privacy or consent not apparent**: We appreciate the feedback. Stack Overflow is released under the CC-BY-SA license. The FreshStack dataset is also released under the same CC-BY-SA license. Licensing details for the GitHub code repositories are provided in Table 7 of the paper.
>
> ### References:
> - [1] FollowIR: Evaluating and Teaching Information Retrieval Models to Follow Instructions. O Weller et al.  NAACL 2025.
> - [2] Overview of the TREC 2020 deep learning track. N Craswell. 2021.
> - [3] Overview of the TREC 2004 Robust Retrieval Track. EM Voorhees. 2004.
> - [4] Support Evaluation for the TREC 2024 RAG Track: Comparing Human versus LLM Judges. N Thakur et al. SIGIR 2025.
> - [5] Overview of TREC 2024 Biomedical Generative Retrieval (BioGen) Track. D Gupta. 2024.

---

### Official Review · Reviewer_z7QE · 2025-07-04

**Ethics Flags:** Data privacy, copyright, and consent
**Rating:** 4
**Confidence:** 4

**Summary:**

This paper presents​FreshStack, a novel framework for automatically constructing challenging and up-to-date information retrieval (IR) evaluation benchmarks. FreshStack addresses three critical limitations in current IR research, e..g., Artificially Easy – Many existing benchmarks fail to reflect real-world retrieval difficulty and Static – Traditional datasets quickly become outdated. To demonstrate FreshStack’s effectiveness, the authors built five new datasets​focused on fast-growing, recent, and niche topics, ensuring tasks remain sufficiently difficult. They then evaluated these datasets using state-of-the-art retrieval methods, uncovering key insights, e.g.,Rerankers do not consistently improve first-stage retrieval accuracy and Oracle context significantly enhances LLM-generated answers. These findings provide valuable guidance for future IR research, particularly in dynamic and specialized domains.

**Dataset Code Accessibility:**

No

**Dataset Code Comments:**

While the paper presents interesting contributions and the described datasets appear valuable for the research community, I couldn't locate any information about the availability of the code implementation or the datasets themselves. This omission significantly limits the reproducibility and practical utility of the work.

**Ethical Comments:**

The data is from StackOverflow, which may face the data privacy and copyright problem.

**Ethical Considerations:**

Yes, there are ethics concerns that require attention by the authors

**Final Justification:**

This paper has solved most of my concerns and I'd like to raise my score and accept this paper.

**Limitations Weaknesses:**

1, While the paper identifies dataset staleness as a limitation of current IR research and attempts to address this by incorporating temporally diverse data, it fails to provide empirical evidence demonstrating the actual impact of temporal factors on retrieval performance. Specifically, the work does not investigate whether modern IR systems exhibit degraded performance when applied to older data No experiments are conducted to quantify the relationship between data freshness and retrieval effectiveness This represents a significant methodological gap, as the claimed benefits of temporal diversity remain unsubstantiated by experimental results.

2,While the paper provides valuable insights through its comprehensive analysis of different information retrieval methods across the five datasets, the evaluation of fusion approaches exhibits some  limitation regarding generator selection. Specifically, the study focuses exclusively on state-of-the-art generators (GPT-4o and GPT-4.1) for fusion IR methods, which may present an overly optimistic view of performance.Notably absent from the evaluation are weaker but more practically relevant generators like LLaMA-7B

3, I was unable to locate any links to the code or datasets associated with this paper. The availability of these resources is crucial for verifying the reported results and enabling further research in this area.Should the authors make the code and datasets publicly accessible, I would reconsider my evaluation of the paper.

**Strengths Contributions:**

This paper makes a valuable contribution by tackling key limitations in current information retrieval datasets, e.g., their frequent lack of realistic query scenarios. The authors' introduction of five novel datasets represents a meaningful advancement for the IR research community, providing more authentic evaluation environments. Through comprehensive analysis of these datasets, the paper yields several important insights that could guide future research directions. The datasets themselves appear well-constructed to address current shortcomings, offering researchers more robust tools for developing and testing retrieval systems.

---

> ### Author Rebuttal · Authors · 2025-07-31
>
> We appreciate that reviewer z7QE finds our benchmark to be a valuable contribution that tackles key limitations in current IR datasets. We have reviewed the weaknesses associated with our paper and addressed all concerns below:
>
> 1. **Limitations of dataset staleness**: We agree. Although one option is to expand the data to cover more categories, we have opted to instead expand our coverage of model categories. We are observing that recent embedding models (such as Qwen-3-8B [1], Stella-1.5B v5 [2]) perform competitively to the fusion baseline and better than embedding models used in the paper, showcasing that modern IR systems are improving on FreshStack. We provide our updated retrieval results on FreshStack below, and we will add these results to the camera-ready version of the paper.
>
> | | LangChain | LangChain | LangChain | Yolo v7 & v8 | Yolo v7 & v8 | Yolo v7 & v8 | Laravel 10 & 11 | Laravel 10 & 11 | Laravel 10 & 11 | Angular 16, 17 & 18 | Angular 16, 17 & 18| Angular 16, 17 & 18| Godot4 | Godot4 | Godot4 |
> | ----------------------- | ----- | -----  | ------ | ----  | ------ | -----  | -----   | ---- | ---- | -----   | ---- | ---- | --- | -----     | -----
> | **Model Name**        | **αN@10** | **C@20**   | **R@50**  | **αN@10** | **C@20**   | **R@50** | **αN@10** | **C@20**   | **R@50** | **αN@10** | **C@20**   | **R@50**  | **αN@10** | **C@20**   | **R@50** |
> | |
> | Qwen3-8B (Emb)          | 0.331 | 0.694  | 0.423  | 0.393 | 0.728  | 0.567  | 0.421   | 0.748| 0.615| 0.373   | 0.700| 0.502| 0.307 | 0.576   | 0.521   |
> | Qwen3-4B (Emb)          | 0.320 | 0.675  | 0.415  | 0.404 | 0.744  | 0.550  | 0.402   | 0.748| 0.604| 0.304   | 0.618| 0.442| 0.303 | 0.496   | 0.440   |
> | Fusion (4 models)       | 0.337 | 0.700  | 0.477  | 0.304 | 0.627  | 0.534  | 0.425   | 0.748| 0.646| 0.385   | 0.719| 0.532| 0.265 | 0.550   | 0.505   |
> | Stella-1.5B v5          | 0.315 | 0.660  | 0.388  | 0.334 | 0.624  | 0.559  | 0.370   | 0.681| 0.590| 0.330   | 0.630| 0.414| 0.237 | 0.481   | 0.443   |
> | Voyage Large 2          | 0.246 | 0.528  | 0.308  | 0.270 | 0.570  | 0.453  | 0.345   | 0.701| 0.543| 0.304   | 0.625| 0.427| 0.282 | 0.522   | 0.458   |
> | Jina V4 (Emb)           | 0.277 | 0.596  | 0.379  | 0.311 | 0.692  | 0.524  | 0.324   | 0.677| 0.552| 0.279   | 0.539| 0.321| 0.220 | 0.416   | 0.351   |
> | Stella-400M v5          | 0.285 | 0.608  | 0.356  | 0.241 | 0.538  | 0.447  | 0.320   | 0.648| 0.534| 0.288   | 0.619| 0.359| 0.244 | 0.476   | 0.412   |
> | BGE (Gemma-2)           | 0.216 | 0.548  | 0.337  | 0.258 | 0.547  | 0.430  | 0.348   | 0.699| 0.574| 0.323   | 0.571| 0.378| 0.200 | 0.479   | 0.419   |
> | Qwen3-0.6B (Emb)        | 0.259 | 0.588  | 0.369  | 0.260 | 0.504  | 0.383  | 0.288   | 0.593| 0.463| 0.253   | 0.535| 0.356| 0.249 | 0.495   | 0.400   |
> | E5 (Mistral-7B)         | 0.304 | 0.654  | 0.393  | 0.243 | 0.552  | 0.394  | 0.250   | 0.565| 0.470| 0.262   | 0.548| 0.368| 0.217 | 0.444   | 0.359   |
> | text-embedding-3-large  | 0.207 | 0.507  | 0.292  | 0.275 | 0.585  | 0.412  | 0.298   | 0.627| 0.494| 0.271   | 0.556| 0.353| 0.187 | 0.409   | 0.316   |
> | Jina V3 (Emb)           | 0.223 | 0.533  | 0.299  | 0.188 | 0.448  | 0.338  | 0.309   | 0.654| 0.489| 0.224   | 0.536| 0.293| 0.190 | 0.405   | 0.301   |
> | GTE (large) v1.5        | 0.206 | 0.470  | 0.252  | 0.195 | 0.445  | 0.271  | 0.318   | 0.626| 0.482| 0.284   | 0.578| 0.343| 0.127 | 0.348   | 0.240   |
> | BM25                    | 0.230 | 0.475  | 0.261  | 0.137 | 0.342  | 0.337  | 0.319   | 0.602| 0.441| 0.259   | 0.551| 0.340| 0.144 | 0.268   | 0.200   |
> | Nomic Embed (Code)      | 0.224 | 0.518  | 0.292  | 0.227 | 0.539  | 0.390  | 0.222   | 0.532| 0.407| 0.237   | 0.511| 0.356| 0.178 | 0.341   | 0.295   |
> | text-embedding-3-small  | 0.213 | 0.523  | 0.283  | 0.172 | 0.423  | 0.303  | 0.245   | 0.571| 0.438| 0.214   | 0.491| 0.295| 0.197 | 0.392   | 0.330   |
> | CodeRankEmbed           | 0.099 | 0.271  | 0.128  | 0.075 | 0.215  | 0.128  | 0.108   | 0.324| 0.225| 0.146   | 0.363| 0.167| 0.091 | 0.224   | 0.160   |
>
> 2. **Dataset and code availability**: The FreshStack datasets were made publicly available on HuggingFace at the time of submission, and they are linked below the title on the first page. The data includes the filtered questions and answers, nuggets, nugget-level binary document judgments, and parsed document chunks for retrieval. The code will be included in the camera-ready version and provides the pipeline for generating new FreshStack splits, evaluation scripts for dense retriever and multi-vector (ColBERT) models, and custom evaluation metrics.
>
> 3. **Study focuses exclusively on state-of-the-art generators (GPT-4o and GPT-4.1) for fusion IR methods**: There is a confusion here, the generators are not used in fusion IR methods. We want to clarify that both parts are independent. Fusion IR conducts an ensemble fusion of different retrieval techniques and models (L178–179). Generators are used after the retrieval stage (e.g., fusion), to generate a system response to answer the question, given the context of the top 20 retrieved documents from fusion retrieval (L310–314).
>
> Now, coming to why we chose state-of-the-art generators for RAG evaluation, there are three reasons:
> - The use of strong models for answer generation is two-fold: (a) they establish stronger baselines and (b) much higher oracle results.
> - The closed-book answers of strong LLMs are important to evaluate on FreshStack for possible dataset contamination (Table 5). GPT-4.1 has a more recent knowledge cutoff date (June 2024), making for an interesting comparison to GPT-4o (October 2023 cutoff)
> - The usage of strong generators also aligns with existing works, such as BRIGHT [3], that use strong LLMs for query expansion in their work.
>
> Lastly, a reminder that FreshStack **stands on its own** without even the generator component for information retrieval evaluation, e.g., Table 4.
>
> 4. **Data privacy or consent not apparent**: We appreciate the feedback. Stack Overflow is released under the CC-BY-SA license. The FreshStack dataset is also released under the same CC-BY-SA license. Licensing details for the Github code repositories are provided in Table 7 of the paper.
>
> ### References:
> - [1] Qwen3 Embedding: Advancing Text Embedding and Reranking Through Foundation Models. June 5th 2025.
> - [2] Jasper and Stella: distillation of SOTA embedding models. D Zhang et al. 2025.
> - [3] BRIGHT: A Realistic and Challenging Benchmark for Reasoning-Intensive Retrieval. H SU et al. ICLR 2025.

---

> > ### Comment · Reviewer_z7QE · 2025-08-06
> > **reply**
> >
> > Thanks for your rebuttal. It have solved most of my concerns and I'd like to raise my score.

---

### Official Review · Reviewer_wAR3 · 2025-07-08

**Rating:** 4
**Confidence:** 3

**Summary:**

The paper introduces FreshStack, a framework for automatically constructing realistic IR/RAG evaluation benchmarks, with a focus on technical domains. Using FreshStack, the authors source community-asked questions and answers from Stack Overflow, gather relevant technical documents from sources like GitHub, and employ GPT-4o to generate factual "nuggets" and assess document relevance. Evaluation on the constructed benchmark with multiple retriers/rerankers suggests that existing system demonstrate significant headroom for improvement in real-world retrieval tasks.

**Additional Feedback:**

Multiple sentences appear to be in weird wordings. Fo example:
```Line 218-219```: "*This is used to construct standard test collections, such as BEIR and TREC datasets such as the Deep Learning (DL) track"*; ```Line 344 - 345```: "*We likely suspect this as a possibility for GPT-4.1 (especially, mini) surprisingly high effectiveness of the closed-book answer."*

**Dataset Code Accessibility:**

Yes

**Ethical Considerations:**

No, there are no or only very minor ethics concerns

**Final Justification:**

Without experimental evidence of FreshStack being applied to other domains (e.g., biomedical) to support its generalization claims, I maintain my score, which reflects the current state of the manuscript. I would be supportive of acceptance if others advocate strongly.

**Limitations Weaknesses:**

- While the authors present FreshStack as a general framework and claimed to be applicable across domains (```Lines 65–66```), the current evaluation is limited to a single setting: technical Q&A data. Moreover, the quality of FreshStack’s automatic pipeline is assessed on only one topic by a single individual (an ML expert). As a result, the claim of generalizability might require further justification. It would be helpful to provide concrete elaboration of the types of structures or domains to which FreshStack could potentially generalize.

- The paper leverages GPT-4o to conduct nugget-level binary relevance judgments across all pooled document chunks. While the authors argue that this approach is more efficient than traditional human annotation, they provide no analysis of its computational cost. Given that technical domain questions tend to be long and contain multiple nuggets, the expense of running large-scale inference could be prohibitively high. It would be helpful to include statistics on the cost of constructing the benchmark with FreshStack, especially for those considering adopting it to build new IR evaluation datasets.

- Many terms, such as nugget generation, nugget-level support, ensemble fusion, and pooling, are central to the proposed method, but they appear without clear definitions or explanations in the Introduction. Including brief descriptions would make the paper more accessible to readers unfamiliar with RAG/IR literature.

**Strengths Contributions:**

- The paper addresses the growing need for realistic and challenging evaluation benchmarks for retrieval and RAG systems. FreshStack effectively fills this gap by focusing on recent, niche technical domains and explicitly addressing three major limitations of existing benchmarks: the lack of realistic, open-ended user queries; the dominance of artificially easy tasks; and the reliance on static, potentially outdated datasets. This makes the work both timely and relevant for the field.

- The paper is clearly organized, making it easy to read and understand. The logical presentation of the findings and content also enhances clarity and aids reader comprehension.

- The experimental setup is thoroughly detailed, covering dataset statistics, model configurations, and evaluation procedures, which promotes reproducibility and reinforces the study’s empirical results.

---

> ### Author Rebuttal · Authors · 2025-07-31
>
> Thank you for your insightful feedback and for recognizing FreshStack as a challenging retrieval benchmark, an organized paper, and a thorough experimental setup. The concerns raised require more explanation, and we have addressed each of them below:
>
> 1. **FreshStack generalization: key assumptions and examples.**
> While FreshStack can be generally applied to any domain, we applied it in a single setting covering technical Q&A data. The key assumptions for FreshStack to generalize to a domain are:
>
> - a. requires at least 50 queries*** that are human-generated for a lower variance in retrieval performance
> - b. answers to the queries (preferably human-generated) that can be linked using information available in documents
> - c. a collection of at least 5-10K documents (or chunks) with rich information
>
> Now applying these assumptions, for FreshStack to generalize to the bio-medical domain, we require these: (1) users writing questions for which they need answers, (2) a bio-medical expert providing an ideal answer, a one-paragraph text that answers the question in a manner that the expert finds satisfactory. (3) a corpus containing all scientific articles from PubMed that can be linked to the answer. We will add these key assumptions and examples as a discussion section in the camera-ready version.
>
> ***50 is the typical minimum number of queries used in TREC. TREC Deep Learning (DL) track 2020 used 54 queries [1]. TREC is a prominent conference for creating information retrieval datasets, and has been running for 30+ years.
>
> 2. **The cost of creating FreshStack using our pipeline is approximately 260–275 USD**: Here is a rough estimate of our costs in constructing FreshStack: (a) **nugget generation**: (O(($\alpha_i$ + $\alpha_o$) * n) cost, where n is the number of queries) costs about 1015 USD with GPT-4o (2024-08-06) to generate nuggets for all domains (2–3 USD per domain). (b) **nugget-level support**: (O(($\beta_i$ + $\beta_o$) * n * m) cost, where m is the depth of retrieved documents for each query) costs about 250–260 USD with GPT-4o (2024-08-06) to assess document support with query nuggets (40–75 USD per domain). So, overall, our computational cost is around 260–275 USD. The $\alpha$ and $\beta$ are constants mapping to expected token count, with subscripts i and o corresponding to input and output. We utilized GPT-4o, but it can now be replaced by cheaper alternatives, like GPT-4.1 variants (mini is quite good) or even Gemini alternatives such as Gemini 2.5 Flash-Lite [2], which are effectively 10x cheaper than GPT-4o.
>
> - **The cost of manually annotating FreshStack would have been approximately 19–20K USD**: Assuming at least 50 documents are labelled for each query, this leads to 57,450 pairwise comparisons (query-document pairs). Assuming a single human annotator can annotate 1 pair in a minute (read the query nuggets and document carefully and mark which nuggets are sufficiently supported by the document), completing all judgments would require approximately 950–1000 hours. Assuming annotators are paid 20 USD / hour, the cost of manually annotating FreshStack is approximately 19–20K USD.
>
> 3. **Clear definitions for terms will be added to the camera-ready version**: Here are clear definitions for terms you have highlighted in your review. We apologize for not making these clear in the draft: (i) **nugget generation**: The nugget-generation methodology was coined two decades ago in the TREC-QA 2001 track for evaluating answers to free-form questions [3]. Human annotators would manually write “information nuggets” or atomic facts present in the answer. More recently, in Pradeep et al. [4], automatic nugget generation is explored, i.e., using LLMs to automatically create nuggets, used for RAG evaluation.
> (ii) **nugget-level support:** This term is analogous to relevance judgment in a traditional IR setting, but we effectively coined it “support” as we calculate whether the retrieved document can sufficiently support the information present within the nugget, instead of relevancy.
> (iii) **ensemble fusion**: Ensemble fusion [5] is defined as the process of combining multiple search techniques (or models) to increase the overall relevance and accuracy of retrieved results.
> (iv) **pooling**: Pooling [6] [7] is a predominant technique used in IR for selecting a subset of documents to be assessed for relevance, instead of assessing every document from the collection.
>
> 4. **Multiple sentences with weird wordings**: We will go through the draft thoroughly to make sure sentences with weird wordings are rewritten properly. Thanks!
>
> ### References:
> - [1] Overview of the TREC 2020 deep learning track. N Craswell. 2021.
> - [2] Gemini 2.5 Flash-Lite. Blogpost 2025.
> - [3] Overview of the TREC 2001 question answering track. EM Voorhees. TREC 2001.
> - [4] Initial Nugget Evaluation Results for the TREC 2024 RAG Track with the AutoNuggetizer Framework. R Pradeep et al. SIGIR 2025.
> - [5] Fusion in Information Retrieval: SIGIR 2018 Half-Day Tutorial. O Curland and J Culpepper. SIGIR 2018.
> - [6] Bias and the Limits of Pooling for Large Collections. C Buckley. SIGIR 2006.
> - [7] How Reliable Are the Results of Large-Scale Information Retrieval Experiments? J Zobel. SIGIR 1998.

---

> > ### Comment · Reviewer_wAR3 · 2025-08-07
> >
> > Thank you to the authors for their detailed responses, which address some of my concerns. Without seeing FreshStack actually applied in action to other domains (e.g., biomedical) through experiments, I believe my current rating reflects the current state of the manuscript. Therefore, I maintain my score but would be supportive of acceptance if others advocate strongly.

---

### Note · Authors · 2025-08-12

Thank you for the constructive feedback. We have incorporated your comments, which have further clarified and strengthened our submission. We appreciate the recognition of FreshStack’s key contributions, in particular:

1. FreshStack is a **realistic and timely** IR & RAG benchmark
2. The submission emphasizes **reproducibility**, with the dataset & code released
3. The empirical results and analysis provide **many insights**, e.g., fusion/oracle behaviors and headroom for retrievers

We hope our work will facilitate future RAG research, given FreshStack's nugget-level and answer evaluation that enables a diverse and realistic retrieval assessment while being inexpensive and generalizable.

Below, we summarize how we addressed the main concerns:

- **Cost & scalability**: We added a concrete cost breakdown, constructing FreshStack via our GPT-4o pipeline is 260–275 USD (nugget generation is 10–15 USD; nugget-level support is 250–260 USD), versus an estimated 19–20K USD for equivalent manual annotation. We note that more affordable, high-performing models (e.g., Gemini 2.5 Flash-Lite) are available for future deployments.

- **Ethics, licensing, and PII**: We clarified licensing: FreshStack is released under CC-BY-SA (matching Stack Overflow). Although the GitHub code originated from sources with mixed licenses, CC-BY-SA is a copyleft license, so we deemed it appropriate to apply it to the entire FreshStack dataset. The individual GitHub repo licenses are in Table 7, and our dataset documentation currently states: "The original GitHub repositories used for constructing the corpus may contain non-permissive licenses; we advise the reader to check the licenses for each repository carefully". We will move the discussion on data licensing and contamination into the main paper. Additionally, we will clarify that we did not perform any PII-specific filtering beyond what is already done by our data providers.

- **Generalizability claims**: The paper already compares models with two different knowledge cut-offs (GPT-4o, GPT-4.1). We have now added results for 13 more embedding models, spanning older (e.g., GTE-large) to very recent (e.g., Qwen-3, Jina-v4), to analyze the freshness effects. The pipeline has been tested across five domains; we also include key assumptions for its use and a hypothetical biomedical application to illustrate adaptability to new domains. Finally, we provide qualitative failure-mode examples to accompany the statistics in Table 3.

---

### Decision · Program_Chairs · 2025-09-18

**Decision:**

Accept (poster)

**Comment:**

The paper proposes a framework to construct realistic benchmarks for IR and RAG. It includes several steps from document collection, nuget generation, to nuget support retrieval by fusing retrieval approaches. The paper addresses an important problem in IR evaluation - developing realistic, large scale evaluation datasets.

Strength: The methodology used in the paper is innovative, although it is built upon some similar existing approaches. The dataset is valuable. The experimental results demonstrate that there is large room for improvement using the current approaches.

Weakness: The reviewers mentioned the problem of licences and data privacy, the limited domains in the datasets, as well as some unclear terminologies.

Discussions: The authors have provided answers to the reviewers' comments. Ethic evaluations are added, recommending the authors to include clearer statements about licence and data privacy, which the authors promised to do. The unclear terminologies are due to the difference in communities - they are widely used in IR experiments (TREC, for example). So some explanations as agreed by the authors will solve the problem. The authors also explained the generalizability problem, and how the framework can be applied to a different domain. The answer is satisfactory.

The paper is not limited by providing a new dataset. It also provides a new methodology for its construction. The approach is innovative and may inspire more work in the same direction. I recommend to accept the paper.